

# Surface area and Ω-aragonite oversaturation as controls of the runaway precipitation process in ocean alkalinity enhancement

Niels Suitner[1], Jens Hartmann[1], Selene Varliero[2], Giulia Faucher[3], Philipp Suessle[3], and Charly A. Moras[1]

[1]Institute for Geology, University of Hamburg, Bundesstrasse 55, 20146 Hamburg, Germany

[2]Department of Chemistry, Materials and Chemical Engineering "Giulio Natta", Politecnico di Milano, Milan; Italy

[3]GEOMAR Helmholtz Centre for Ocean Research Kiel, Wischhofstrasse 1-3, 24148 Kiel, Germany

Correspondence: Niels Suitner     (niels.suitner@uni-hamburg.de),

      Jens Hartmann     (geo@hattes.de)

**Orcid:**

Niels Suitner:   https://orcid.org/0000-0003-3413-857X

Jens Hartmann:   https://orcid.org/0000-0003-1878-9321

Selene Varliero:   https://orcid.org/0000-0001-9532-2202

Giulia Faucher:   https://orcid.org/0000-0001-8930-477X

Philipp Suessle:   https://orcid.org/0000-0001-7224-2330

Charly A. Moras:   https://orcid.org/0000-0001-6819-6167



**Abstract**
Ocean alkalinity enhancement (OAE) is a strategy for marine carbon dioxide removal that aims to increase the
total alkalinity (TA) of seawater to sequester atmospheric $CO_2$ in the form of dissolved inorganic carbon (DIC).
An intense alkalinization of seawater resulting from OAE treatment could trigger a significant runaway carbonate
precipitation process, which may lead to a loss of initially added alkalinity, thereby limiting its efficiency. Even
under natural background aragonite saturation states, a continuous yet barely detectable loss of alkalinity is
theoretically expected to occur in seawater. With the additional increase through OAE, time ranges to initiate an
appreciable TA loss process could be reduced significantly. Therefore, predicting the alkalinity stability ranges
might be a necessity for application scenarios. The main drivers of the precipitation process are i) the aragonite
saturation state of seawater and ii) the available surface area for heterogeneous precipitation.
In this study, we refined the use of logistic functions to describe the temporal evolution of both drivers, with
experimental datasets using natural seawater from the Raunefjorden (Bergen, Norway; Temp.: ~11°C, Sal.: ~32.6).
The observed patterns were then used to derive a process-based model for calculating TA-loss rates, focusing on
the accelerated precipitation phase of the runaway process while considering saturation levels and available
particle surface area. The formation of carbonate phases reduces seawater TA concentrations, inducing a delay or
stopping the TA-loss process. In addition, the sinking of precipitated particles decreases the potential for further
precipitation by reducing the available surface area in the system. To assess the impact of particle sinking on TA-
loss, their shape and size distribution were determined. Under the environmental conditions presented here, TA-
loss rates could be reduced by up to 30-40% due to the sinking of particles, after just one day.
Integrating the proposed concepts into ocean models could enhance the accuracy of predictions regarding the fate
of added alkalinity. Gaining insights into the evolution of the identified, seemingly stable TA levels can help
prevent accelerated precipitation phases. Additionally, an understanding of particle sinking or dilution processes
reducing the available reactive particle surface area is relevant to assess the efficacy and durability of OAE.





## 1 Introduction

To mitigate climate change and reach net-zero greenhouse gas emissions by the end of the century, negative emission technologies (NETs) are necessary (UNFCCC, 2015) considering the slow change in the development of the energy infrastructure, lifestyle of humanity, and national goals for economic growth (Fuss et al., 2018; Iyer et al., 2015; Sers & Victor, 2018). Various carbon dioxide removal (CDR) technologies have been proposed to help achieve the necessary negative emission trajectories (Hartmann et al., 2013; IPCC, 2023; Minx et al., 2018; Rogelj et al., 2018). Among these, ocean alkalinity enhancement (OAE) is a promising CDR method (Harvey, 2008; Ilyina et al., 2013; Kheshgi, 1995; Rau & Caldeira, 1999), with the potential to geochemically sequester 3-30 Gt $CO_2$ yr$^{-1}$ (Oschlies et al., 2023; Renforth & Henderson, 2017).

Alkalinity enhancement could be achieved by two addition approaches: **1.** a non-$CO_2$-equilibrated (neq) or **2.** a $CO_2$-equilibrated (eq) (Schulz et al., 2023). Through the neq approach, alkaline materials, such as silicate or hydroxide-based mineral phases could be introduced to seawater in the form of solids or solutions, allowing longer-term $CO_2$ equilibration with the atmosphere through ingassing of atmospheric $CO_2$. In the eq approach, already partially pre-$CO_2$-equilibrated solutions or carbonate-based substances could be released into seawater. Neq alkalinity addition strategies induce greater variations in the affected carbonate system, resulting in drastically reduced $pCO_2$ and a rapid increase in pH values. While an eq alkalinity addition results in less severe changes in ocean chemistry, it is less efficient in generating carbon sequestration potential (Schulz et al., 2023; Suitner et al., 2024).

Depending on the introduced alkalinization method (see Eisaman et al., 2023) and the magnitude of treatment, induced changes in the carbonate system could lead to adverse effects on biota (Faucher et al., 2024; Ferderer et al., 2022; Gately et al., 2023; Goldenberg et al., 2024; Marín-Samper et al., 2024; Sánchez et al., 2024; Xin et al., 2024) or in case of persistent oversaturation, result in the precipitation of secondary mineral phases and therefore a loss of the introduced total alkalinity (TA) (Iyina et al., 2013; Schulz et al., 2023). The process of TA leakage as a consequence of OAE was recently described by several studies (see Fuhr et al., 2022; Hartmann et al., 2023; Moras et al., 2022; Pan et al., 2021; Ringham et al., 2024; Suitner et al., 2024; Varliero et al., 2024). Within these laboratory-based studies, self-sustaining runaway carbonate precipitation processes led to a significant decrease in the added alkalinity, which could even result in a net-loss of TA. TA stability ranges, and the evolution of the precipitation process depend on the specific local environmental conditions such as temperature, salinity, aragonite saturation state ($\Omega_{ar}$), or suspended particle load of the treated water mass (Moras et al., 2024).

The objective of this study is to demonstrate the general capability to predict and parametrize the temporal evolution of a triggered runaway carbonate precipitation process during OAE approaches, based on quantifiable and measurable parameters. Estimations of stability ranges for the permanence of introduced alkalinity additions were derived from these parametrizations. The ability to predict TA stability ranges can help prevent secondary mineral formation and optimize assessments for future OAE application scenarios.

Suitner et al. (2024) demonstrated the potential of utilizing inverse logistic functions to depict the temporal evolution of the TA-loss process during the runaway carbonate formation phase (see Fig. 1). In this study, principal descriptive parameters such as TA addition and stability ranges to trigger the runaway process or the timespan of the precipitation phase could be formalized based on their experimental dataset. This approach also offers the possibility of a straightforward integration of time-dependent loss terms into predictive computational models



simulating OAE addition scenarios, as presented by He & Tyka (2023), Ou et al. (2025), Schwinger et al. (2024)
or Zhou et al. (2024).
The application of OAE may exceed critical levels for carbonate precipitation. For open ocean scenarios the rapid
initiation of mixing processes would efficiently reduce the potential for secondary carbonate formation.
Nevertheless, runaway carbonate formation may occur within enclosed geographic locations with physical features
such as bays, estuaries, or lagoons. In addition, thermohaline layering (Bialik et al., 2022) or high sediment load
(Wurgaft et al., 2016, 2021) might create conditions that lead to alkalinity loss processes.
To sustain the observed runaway carbonate formation (Fig. 1), it is essential to retain the precipitates in the system.
Removal of these particles reduces the potential for continuous heterogeneous precipitation, thereby slowing down
or halting the process. In this study, the empirically determined alkalinity loss rates were used to derive the quantity
of precipitated particles. By identifying the particle sizes, shapes, densities, and sinking velocities, their potential
residence times in the water column were estimated. Furthermore, we evaluated whether the formation of
secondary minerals can supply sufficient surface area for a continuous detectable heterogeneous runaway process
in an open-water body and whether the process would be interrupted or attenuated by removing particles due to
their descent into deeper layers.

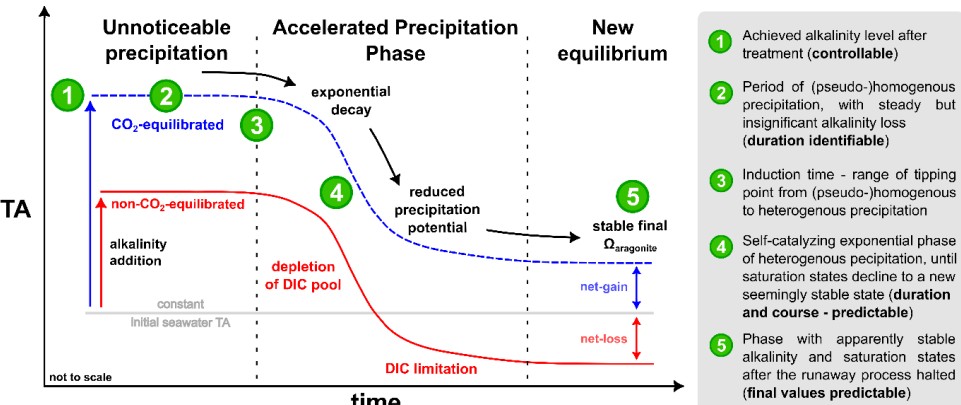

*Figure 1: Modified conceptual scheme of a runaway carbonate precipitation process following liquid alkalinity addition after Fig. 9 in Suitner et al., 2024 (not to scale)*




## 2 Material and methods

### 2.1 Overview of experimental setups

Three incubation experiments were conducted to examine the stability of alkalinity of the local "filtered" (mesh size 0.2 µm) and "unfiltered" (mesh size 50 µm) seawater of the Raunefjorden, Norway (60.27° N, 5.20° E). Within TA-gradient approaches, runaway precipitation was observed in eq and neq treatments, after surpassing specific time and alkalinity addition ranges, allowing the description of patterns during the precipitation process. A detailed description of the experimental results, design and methods is already given in Suitner et al. (2024), a brief overview is also provided in Tab. 1.

*Table 1: Overview of the experimental design of precipitation experiments in Suitner et al. (2024)*

| # | filter mesh size | $CO_2$ state to atmosphere | Alkaline material | Runtime [days] | Range $TA_{added}$ [µmol kg$^{-1}$] | $TA_{added}$ gradient steps [µmol kg$^{-1}$] | Temperature [°C] |
|---|---|---|---|---|---|---|---|
| I | 50 µm | non-equilibrated (neq) | NaOH | 25 | 0-2800 | 200 | 10-11 |
| II | 0.2 µm | | | | 0-3400 | | 11-13 |
| III | 0.2 µm | air-equilibrated (eq) | $Na_2CO_3$/ $NaHCO_3$ | 20 | 0-9200 | 800 | 12-16 |

### 2.2 Curve fitting of the TA and $\Omega_{ar}$ evolution

The numerical curve fit model to describe the temporal development of TA and $\Omega_{ar}$, as presented in Suitner et al. (2024), was refined by additionally including the observed TA-loss rates as a second input factor, to provide continuous functions as a basis for further model calculations. The curve fit model utilized the consistent tendency of all observed runaway precipitation processes to follow inverse logistic trends in form of:

$$f(t) = a\,e^{-b\,e^{-c\,t}} + d \qquad\qquad (1)$$

for the temporal evolution of TA and $\Omega_{ar}$. The coefficients **(d)** and **(a)** are defined by the achieved level of TA/$\Omega_{ar}$ after the addition **(d)** and the final reached value after the runaway process halted **(a)**. Since these factors are predefined by the experimental setup, the curve fit model only numerically parameterizes the two coefficients **(b)** and **(c)**. Coefficient **(b)** represents the "induction time", or the time required for $CaCO_3$ precipitation to become detectable in the TA measurements, depicted by the horizontal translation along the x-axis, while **(c)** denotes the timespan between start and end of an accelerated precipitation phase (APP). See Fig. 2 for a visual impression of the influence of iterations of each coefficient.





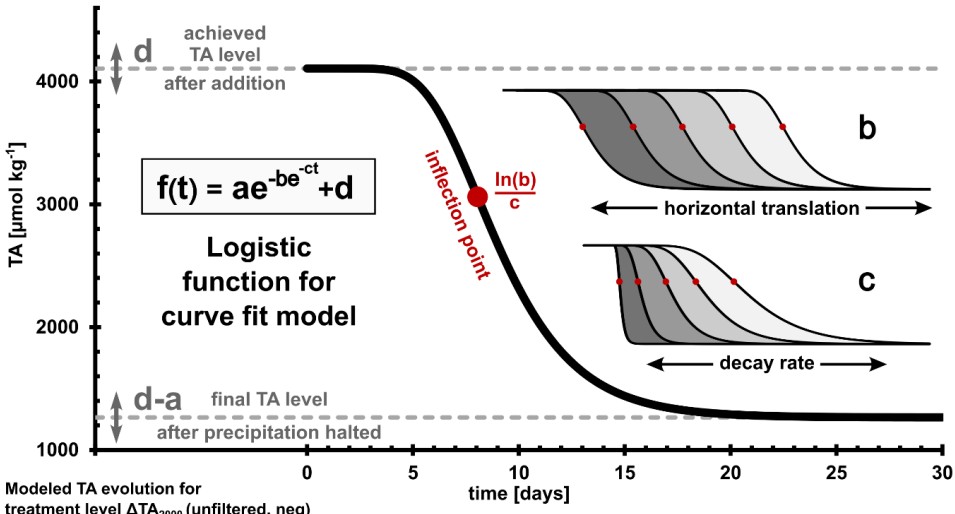

*Figure 2: Overview and iterations of each coefficient (**a**)-(**d**) of the applied inverse logistic function for the numerical curve*

*fitting; the inflection point is defined by ln(b)/c; for further characteristics see Tjørve & Tjørve (2017)*


### 2.3 Empirical rate law and determination of loss rates

A simple empirical rate law was used to evaluate the precipitation rates R [μmol m$^{-2}$ h$^{-1}$] (see e.g. Inskeep &
Bloom, 1985; Morse et al., 2007; Zhong & Mucci, 1989):
$$R = k(\Omega_{ar} - 1)^n \quad (2)$$
The experimental TA-loss rates were then fitted to the logarithmic form of Eq. (2) to determine the coefficients k
(rate constant) and n (empirical reaction order):
$$\log(R) = n(\Omega_{ar} - 1) + \log(k) \quad (3)$$
To correct for the variable surface area, r [μmol h$^{-1}$] was normalized for the assumed available active mineral
surface area (A in [m²]) (adapted from Sjöberg, 1976).
$$r = k A (\Omega_{ar} - 1)^n \quad (4)$$
As the gradient approaches could not provide a sufficient amount of precipitates to determine their surface area, a
one-week side experiment was conducted to estimate the mineral surface area generated during the runaway
precipitation process. By adding 3.8 mmol NaOH and 7.0 mmol NaHCO$_3$ to 40 L of natural seawater (salinity 33)
at 23°C, around 5 g of aragonite precipitates were generated to provide material for a BET surface area
measurement. Using N$_2$ adsorption (Brunauer et al., 1938), with a Quantachrome autosorb iQ at the University of
Hamburg, a surface area of 2.283 m² g$^{-1}$ was determined. By the assumption that the surface area is constant for
all precipitates and that the entire lost TA is transformed into aragonite particles, the experimentally determined
TA-loss was used to calculate the surface area after each timestep, therefore allowing to correct the precipitation
rates.



**2.4 Particle analysis**
The precipitated particles of three filters, collected during incubation experiments within previous campaigns
published in Hartmann et al. (2023) (neq $\Delta TA_{2400}$, Gran Canaria) and Suitner et al. (2024) (neq $\Delta TA_{2600}$ and
$\Delta TA_{2800}$, Raunefjorden), were analyzed by scanning electron microscopy (SEM; Tabletop Microscope Hitachi
TM4000plus - University of Hamburg) to determine shape, size and quantity of the precipitated material. Length,
width and shape of each particle were specified by manual examination.
If sufficient precipitated materials could be provided, the remaining SEM filter material was used to determine
their sinking velocities, utilizing a FlowCam (Fluid Imaging Technologies Inc., Scarborough, United States). A
detailed description of the setup is provided in Suessle et al. (2023) and references therein.



## 3 Results

### 3.1 Numerical logistic curve-fittings

Three OAE gradient approaches by Suitner et al. (2024) were examined to test the stability of alkalinity and to generate refined numerical logistic curve fittings of the temporal development of TA and $\Omega_{ar}$ (Figs. 3, S2 and S3). The coefficients (**b**) and (**c**) (see Fig. 2) were determined by numerical interpolation to optimize the fit to Eq. (1) and its derivative in equal proportions. Therefore, the shown functions were optimized to describe the temporal evolution, while also including the rate loss changes, which allowed an improved description of the runaway process in comparison to the approaches in Suitner et al. (2024). Outlying data points displaying an anomalous increase or stagnation in values were removed from curve-fitting calculations (see SI in Suitner et al., 2024). For each treatment, continuous differentiable functions to describe and analyze the runaway carbonate precipitation process during OAE approaches were generated. To illustrate the described processes and trends, the unfiltered neq approach was selected as an example (Fig. 3). The plots for the filtered approaches are provided in the SI (Figs. S2 and S3).

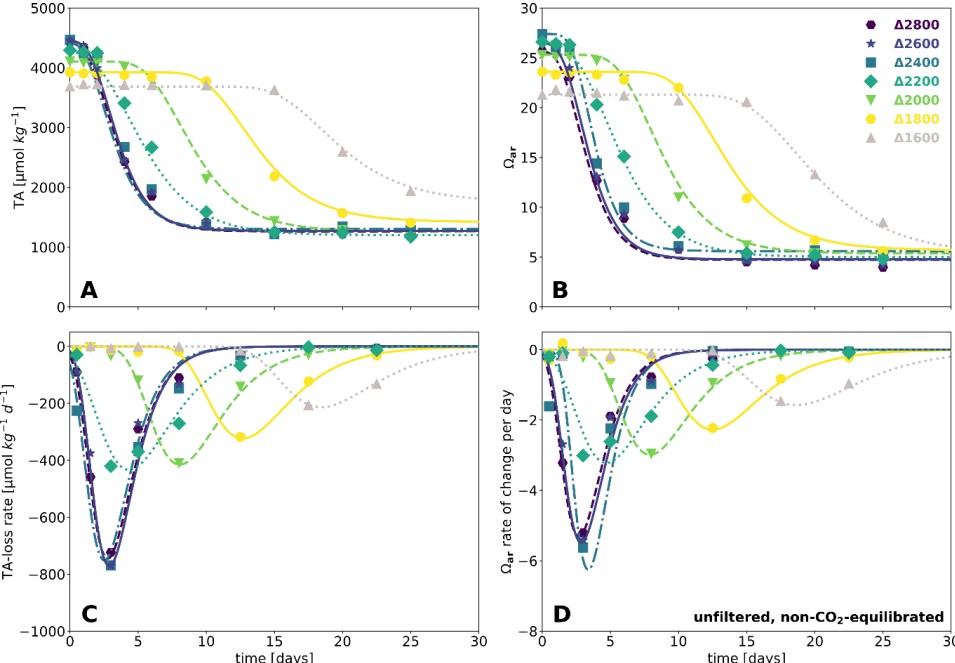

*Figure 3: Results of the numerical curve fits – for the unfiltered neq approach, TA evolution over time (A), $\Omega_{ar}$ evolution over time (B), TA-loss rate over time (C), $\Omega_{ar}$ rate of change over time (D). line plots: curve-fitted continuous functions, markers: measured data points, for related diagrams for filtered approaches see Figs. S2 and S3*

For the unfiltered neq experiment (Fig. 3), treatment levels $\Delta TA_{1600}$ and higher entered into an APP after exceeding critical TA levels to initiate the runaway carbonate precipitation process. Treatments levels $\Delta TA_{2400-2600}$ exhibited a buffering as a consequence of magnesium hydroxide precipitation (see Badjatya et al., 2022; Ringham et al., 2024; Suitner et al., 2024; Varliero et al., 2024), which prevented an increase above ~4470 µmol kg$^{-1}$ in TA and





~27.4 in $\Omega_{ar}$. The buffering effects were not recognized within the fitting procedure and the first data point (after
~3 min runtime) of each treatment level was set as the baseline.

**3.2 Induction time and timespan of the APP**

By employing the logistic curve fits, the temporal evolution of each approach could be parameterized. To identify
the temporal stability ranges and reflect the transition from stable to precipitation-dominated system modes, a
criterion of 40 µmol kg$^{-1}$ d$^{-1}$ rate of change in TA was set. This rate provides a sufficiently high threshold to exclude
a false detection due to natural variability or measurement errors, while still being low enough not to overlook a
significant fraction of alkalinity loss (see Fig. S9 for varying criteria).
Based on this criterion, Fig. 4 illustrates the induction times for the APPs. The shaded ranges indicate extrapolated
timeframes between subsequent measurements during which the initiation of the APP for each treatment was
detected. The displayed regressions were calculated using the averaged times from two consecutive measurement
days. For comparison, hollow markers represent predictions from the presented curve-fitted functions. The
regressions of the induction times uniformly follow an inverse exponential trend of the type:
$$t(TA) = f e^{-g\,TA} - h \tag{5}$$
The employed data series covered a range of 25 days with progressively increasing induction times from 0 to 20
days for treatments reaching ~4470 ($\Delta TA_{2400}$) to ~3380 µmol kg$^{-1}$ ($\Delta TA_{1200}$) in the filtered neq experiment and
~11200 ($\Delta TA_{9200}$) to ~6500 µmol kg$^{-1}$ ($\Delta TA_{4400}$) in the filtered eq experiment. Treatment levels above $\Delta TA_{2400}$ in
the neq approaches exhibited an immediate onset of TA-loss due to the precipitation of secondary hydroxides
and/or carbonate minerals, therefore, following the presented criterion, practically leading to their immediate entry
into the APP process.
The same relationships and trends can also be applied using $\Omega_{ar}$ as a variable. While the neq approaches exhibited
lower $\Omega_{ar}$ values (17.8-27.4) compared to the eq treatments (19.5-43.6), the onset of the APP in the neq experiments
occurred significantly earlier. This indicates that $\Omega_{ar}$ is not the only decisive factor guiding the (pseudo-)
homogeneous nucleation process, determining the induction time.





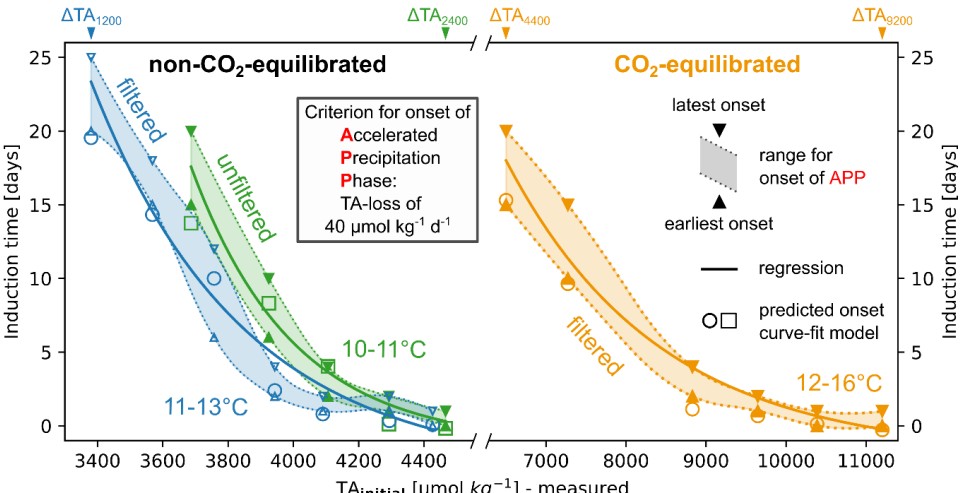

Figure 4: Induction time for the onset of APP in relation to the initial TA addition level, based on the first detection of a TA-loss rate of 40 µmol kg⁻¹ d⁻¹. Each pair of triangle markers represents two consecutive measurement days during which the set loss rate criterion was met; hollow markers: predicted induction times for each treatment level, based on the introduced curve-fit model. Exponential regression of average experimentally detected induction time, see Eq. (5) in Tab. 2 for related functions

Table 2: Regressions of induction times, see Fig. 4. Note that the use of the given equation should not be generalized, as it is only valid under the presented environmental conditions. Also be aware that the resulting predictions of induction times far out of the specified TA addition ranges might not be accurate.

| $t(TA_{initial}) = f\,e^{-g\,TA_{initial}} - h$ (5) | | Regression | | | |
|---|---|---|---|---|---|
| Treatment | | Temp. [°C] | $f * 10^3$ | $g * 10^{-4}$ | h | $R^2$ |
| non-equilibrated | unfiltered | 10-11 | 2721.769 | 32.233 | 1.215 | 0.996 |
| | filtered | 11-13 | 39.633 | 21.646 | 2.972 | 0.977 |
| equilibrated | filtered | 12-16 | 0.603 | 5.243 | 1.934 | 0.988 |

### 3.3 Timespan of APP

To describe the temporal evolution of TA and $\Omega_{ar}$ during the observed runaway processes for the present setups, coefficients (**a**) and (**d**) in Eq. (1) can be set, while (**b**) could be evaluated by empirical or modeled data. Consequently, only the duration of the APP represented by (**c**) needs to be estimated to enable the entire model description of the precipitation procedure. The discrete nature of sampling days with decreasing frequency of samplings towards the end of an experiment (up to 5 days) did not allow reliable empirical determinations of (**c**). The displayed APP timespans in Fig. 5 were therefore determined by the predictions of the presented curve-fits (Fig. 3), based on the 40 µmol kg⁻¹ d⁻¹ TA-loss criterion to define the start and endpoint of the APPs. Fig. 5 illustrates the related predicted timespans against the initially reached TA levels, categorized by the individual experimental setups. The neq APPs form distinct clusters for each approach, which again can be subdivided into treatments with and without the occurrence of immediate precipitation. Regardless of the initial TA enhancement level, treatments that exhibited an immediate decline due to Mg(OH)₂ formation showcased almost identical APP



spans (unfiltered ~8.8 and filtered ~5.9-7.4 days) within each approach. Although the neq treatments without
Mg(OH)$_2$ had the same starting conditions, the unfiltered experiments exhibited approximately 4 days longer
APPs. APPs in the eq approach showed a continuous increase ranging from 5 to 11 days.

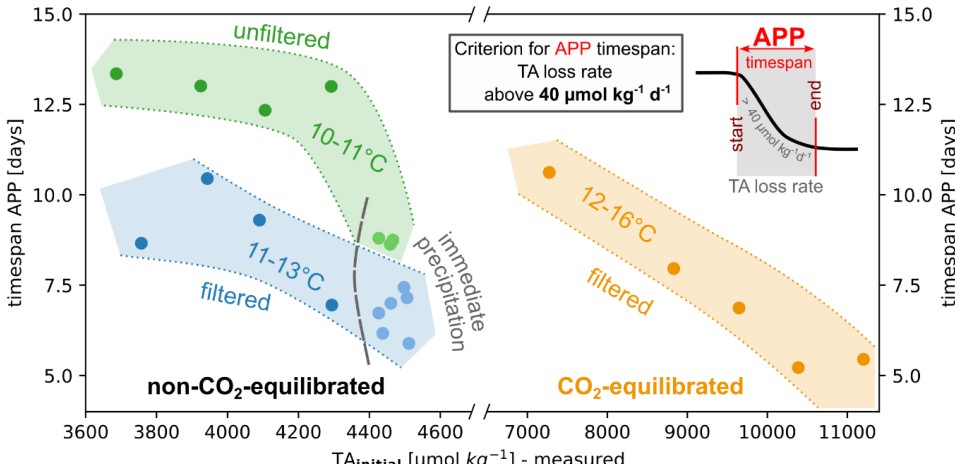

*Figure 5: Overview timespans of APP in relation to the initial TA addition level; determined by the outcomes of the presented numerical logistic curve-fitting. Presented timespans are based on the introduced alkalinity loss criterion (see section 3.2), which was defined as period with rates above 40 μmol kg$^{-1}$ TA-loss d$^{-1}$; only treatments which reached the final stable stage were considered, neq treatments labeled with immediate precipitation showcased a loss of TA within the first 3 min of the experiment – most likely as a consequence of Mg(OH)$_2$ formation (see Suitner et al., 2024)*

## 3.4 Prediction of onset and timespan of APP

The established continuous logistic functions allow estimations of effects occurring between measurement points,
thereby improving the overall accuracy beyond what discrete experimental datasets could provide. Based on the
40 μmol kg$^{-1}$ d$^{-1}$ TA-loss criterion (see section 3.2 and sketch Fig. 5), these functions could therefore assess the
initiation of the APP for specific initial TA or $\Omega_{ar}$ levels (see regressions in Fig. 4). In this regard, Fig. 6 illustrates
the correlations of the curve-fitted coefficients **(b)** and **(c)** and their related entities of the modeled induction times
and APP timespans (see Fig. 2). Under the present physicochemical conditions, the provided regressions could be
utilized as conversion equations to estimate the TA development of a treated water mass based on an existing
database or to convert observational data into mathematically expressible equations for predicting the future
evolution.




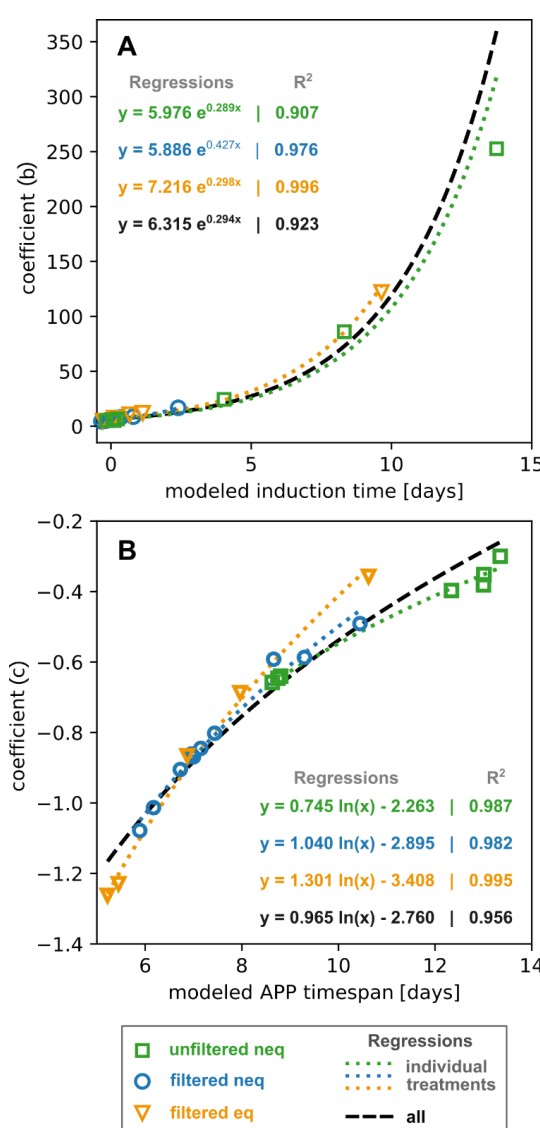

*Figure 6: Regressions describing the relationships between the coefficient (**b**) and the modeled induction time [A], as well as the coefficient (**c**) and the modeled APP timespan [B] for each approach. The shown regressions allow for the conversion of specifiable time-dependent characteristics of a runaway process to the coefficients of the presented inverse logistic function (Eq. (1)). Specified relationships should not be generalized and are only valid within the given conditions of each approach.*

## 3.5 Empirical rate equations

Further implications about the reaction speed and the related timespan of the APP can be provided by empirical rate law equations. As an example, Fig. 7 demonstrates the relationship between the logarithm of TA-loss rates normalized by surface area and aragonite saturation states for the unfiltered neq approach (see Fig. S4 and S5 for details on filtered approaches), for treatments that entered the accelerated precipitation phase. Throughout all experiments log(R) TA-loss rates correlate with the log($\Omega_{ar}$-1), expressing the characteristic relationship for carbonate formation (see Morse et al., 2007; Mucci & Morse, 1983; Zhong & Mucci, 1989). The strong correlation of the linear regressions within each experiment enables the articulation of the empirical rate equations, such as Eq. (2): $R = k(\Omega_{ar} - 1)^n$. In this equation R represents the surface area normalized TA-loss rate, k the rate



constant and n the reaction order. The related values for n and log(k) derived from the linear regressions are
provided in Tab. 3 (see Tab. S1 and S2 for filtered experiments), showing reasonable consistency in n and log(k)
values within each of the three separate experiments. While some treatments, showing immediate $Mg(OH)_2$
formation, slightly deviate, the other treatment levels displayed reaction orders (n) within a relatively narrow range
of 2.45 to 2.73. In comparison, log(k) values ranged between 0.30-1.68, showcasing a higher variability.

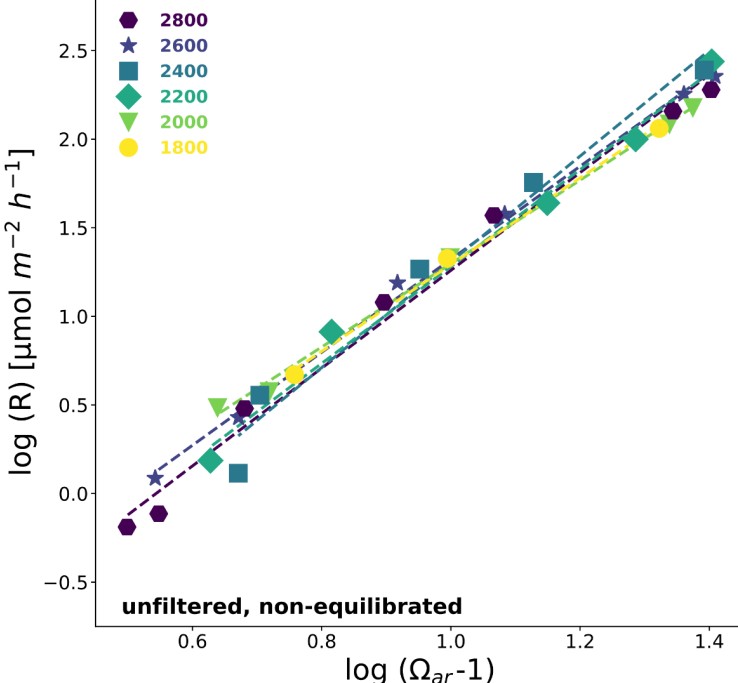

*Figure 7: Carbonate precipitation kinetics for unfiltered neq treatments that entered the APP; see Tab. 3 for related regressions*
*and rate equations.*

*Table 3: Overview of coefficients and regressions of empirical rate equations for unfiltered neq treatments, also see Fig. 7*

| Treatment | $\log(R) = n(\Omega_{ar} - 1) + \log(k)$ | | | |
|---|---|---|---|---|
| ΔTA | n | log(k) | $R^2$ | $\hat{\sigma}$ |
| **2800** | 2.76 | -1.50 | 0.989 | 0.117 |
| **2600** | 2.62 | -1.30 | 0.997 | 0.055 |
| **2400** | 2.98 | -1.68 | 0.975 | 0.167 |
| **2200** | 2.73 | -1.45 | 0.989 | 0.106 |
| **2000** | 2.35 | -1.06 | 0.997 | 0.046 |
| **1800** | 2.45 | -1.16 | 0.996 | 0.060 |
| **all** | 2.68 | -1.39 | 0.985 | 0.106 |




### 3.6 Evolution of particles and sinking speed

To assess the impact of secondary precipitated particles during OAE approaches, precipitated materials from the studies by Hartmann et al. (2023) and Suitner et al. (2024) were analyzed for shape, size, and sinking velocity. As qualitatively depicted in the study of Suitner et al. (2024) (see Fig. 7 therein) the aragonite precipitates manifest and evolve in a variety of forms and sizes, ranging from stem-like structures, followed by double-broccoli shapes and ultimately forming closed spheres.

For this study the length and width distribution of 950 precipitated particles were determined by manual inspection of four overview SEM images (see Fig. S1) from Gran Canaria samples (see Hartmann et al., 2023 for details; analyzed treatment level: $\Delta TA_{2400}$, filtered, neq, Temp. ~23°C, Sal. ~36.5, runtime 4 days) and the Raunefjorden, Bergen (this study and Suitner et al., 2024, see Figs. 3 and 7 therein; analyzed treatment levels: $\Delta TA_{2800}$ and $\Delta TA_{2600}$, Temp. ~11°C, Sal. 32.6, runtime 25 days – highest unfiltered neq treatments). Results of this evaluation are provided in Fig. S6. Length and width distributions of the formed particles follow distinct ratios, allowing the definition of three categories: **1.** Stems (<0.5), **2.** Broccoli (0.5>x>0.9) and **3.** Spheres (>0.9). Note that this method also categorizes regularly shaped, multi-branched particles as spheres (see Fig. 8). Precipitates from the Gran Canarian campaign primarily comprised well-developed broccoli and spherical-shaped particles, whereas the samples from the Raunefjorden were characterized by less evolved stems and broccoli as the dominant components. Although the runtime for the Gran Canarian sample was considerably shorter (4 days) in comparison to the Raunefjorden (25 days), the ~12°C difference in temperature led to significantly higher precipitation rates and more developed shapes. The analyzed Raunefjorden samples originate from the same experiment and differ only in the initially added TA-level of 200 µmol kg$^{-1}$. Even this minor difference in TA addition resulted in the presence of more evolved shapes in the higher treatment.

Based on the distributions of equivalent spherical diameters (ESD), the sinking velocities of the precipitated materials were calculated to identify their hypothetical sinking velocities. To facilitate this calculation, the densities of the aragonite precipitates were determined by actual sinking velocity measurements of the same materials, providing densities of 1.54 to 3.18 g cm$^{-3}$ in an ESD range of 12-50 µm. The discrepancy with the density of aragonite (~2.95 g cm$^{-3}$) may result from an overestimation of particle sizes in the calculation method, which relies on an inversion of Stokes' Law for the terminal sinking velocity of perfect spheres. However, most particles are not spherical and contain numerous cavities within their structure, which likely contributes to an underestimation of particle densities. Therefore, Fig. 8 features a range of sinking velocities of the counted precipitates in dependence to a variable density, supported by ESD distributions and ranges for different types of precipitated particles. Measured sinking velocities for precipitated particles within the aforementioned density range varied from ~5 m d$^{-1}$ (14 µm particle) to ~47 m d$^{-1}$ (41 µm particle). Recorded particles in the ESD range of 50-180 µm were not included in the calculations, as they were not observed within the same filter material that was analyzed by visual inspection, yielding densities of 1.1-1.3 g cm$^{-3}$. Discrepancies between measured and calculated values may reflect aggregation effects for very high values and the technical limitations of the utilized FlowCam to track particles smaller than 3 µm (Bach et al., 2012).

Derived from the calculated sinking velocities, the residence times within the upper 200 m of the water column were determined. Accordingly, early precipitated stages, such as stems (<10µm), could remain for a few months within the upper ocean layer, providing potential additional surfaces for an ongoing heterogeneouss precipitation if a continuous local alkalinization is applied. In contrast, precipitates >30 µm would descend within days to deeper ocean layers, not affecting the precipitation behavior of continuous surface alkalinization attempts. Notice that



291    sinking velocities are temperature and salinity dependent, and therefore would vary under different environmental

292    conditions (see Fig. S7).

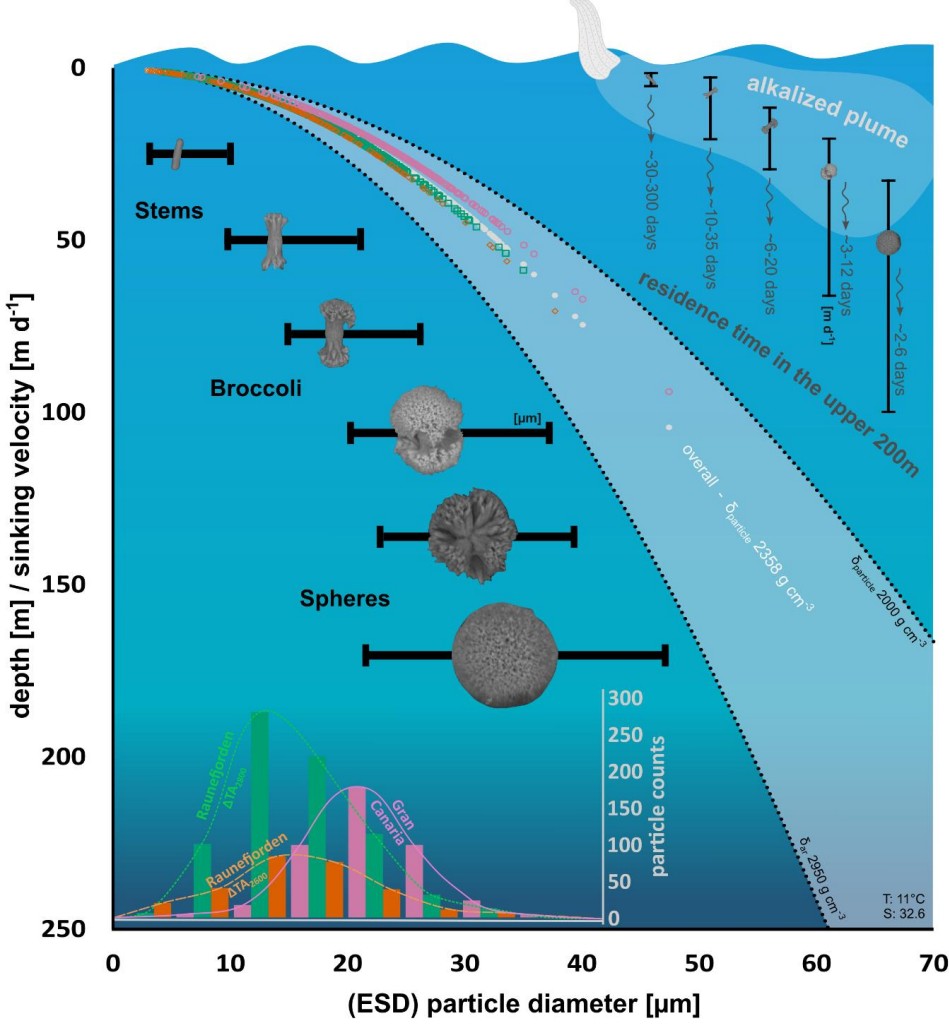

*Figure 8: Overview of grain size distributions of precipitated particles, expressed as ESD (equivalent spherical diameter) in µm (black bars) and particle counts in size fractions steps of 5 µm (histogram, also see Fig. S6); calculated sinking velocities of particles (hollow markers) as a function of ESD for each treatment (green squares: neq unfiltered $\Delta TA_{2800}$ − Raunefjorden, Bergen; orange diamonds: neq unfiltered $\Delta TA_{2600}$ Raunefjorden, Bergen; pink circles: neq filtered $\Delta TA_{2400}$ Gran Canaria). The mean time range for average particles of their class (stems, broccoli, spheres) to sink below the mixed layer depth (assumed to be 200m), neglecting particle growth processes.*

293





## 4 Discussion

### 4.1 General findings

By analyzing the experimental datasets provided by Suitner et al. (2024), this study demonstrates that the process of alkalinity loss during runaway carbonate precipitation follows quantifiable relationships. For the present study, the compiled concepts allowed the description of the principles guiding the entire runaway process. The obtained capability to predict TA-stability ranges, in terms of time and magnitude, might help preventing secondary mineral formation, thereby optimizing the assessments for OAE application scenarios. Furthermore, the simplicity of the logistic curve fit model, along with the demonstration that the carbonate precipitation follows simple rate law equations (see Morse et al., 2007; Mucci & Morse, 1983; Zhong & Mucci, 1989), might facilitate the straightforward integration of these fundamental mechanisms into ocean models like the studies by He & Tyka (2023), Ou et al. (2025), Schwinger et al. (2024), Wang et al. (2022) or Zhou et al. (2024).

### 4.2 Nucleation and onset of accelerated precipitation phase

Previous studies examining the evolution of the runaway precipitation process in the context of OAE (Hartmann et al., 2023; Moras et al., 2022, 2024; Suitner et al., 2024) observed and described considerable periods with stable TA levels before the onset of the APP (e.g. see Fig. 1 modified after Suitner et al., 2024), depending on the alkalinity and DIC levels.

In theory, even at natural background supersaturation levels in the ocean, (pseudo-)homogeneous precipitation is expected to occur at very slow rates, on timescales of thousands of years (Pytkowicz, 1965, 1973). Regardless, the nucleation and precipitation processes in ocean waters are suppressed by inhibitory species like $Mg^{2+}$ (Berner, 1975; Pan et al., 2021; Pokrovsky, 1998), phosphate- (Burton & Walter, 1990) or dissolved organic matter (Chave & Suess, 1970; Kellock et al., 2022; Moras et al., 2024). Naturally occurring precipitation events in the ocean are associated with unique occurrences such as flash floods (Wurgaft et al., 2016, 2021) or whitings (Broecker & Takahashi, 1966; Bustos-Serrano et al., 2009; Morse et al., 2003), providing high degrees of (re)suspended sediments that catalyze a heterogeneous carbonate precipitation procedure.

To consider the persistent (pseudo-)homogeneous precipitation within typical natural seawater supersaturation ranges, the terminology concerning specific stability ranges of TA or timeframes for the onset of secondary carbonate formation should be refined. However, within typical observation times in the Earth system, the precipitation of secondary calcium carbonate in particle-free seawater solutions is expected to be suppressed to $\Omega_{ar}$ values of approximately 11.3 or below (derived from Eq. (4) in Marion et al. (2009), based on data by Morse & He (1993) and Morse et al. (2007)).

Nevertheless, even a 0.2µm-filtered natural seawater contains around $\sim 10^9$ particles per ml in the size range of 5-120 nm, already offering a total surface area of around 8 m² per m³ (cf. Wells & Goldberg, 1992), potentially acting as a catalyst to initiate carbonate precipitation in alkalinity treated seawater. In the presence of surfaces for pseudo-homogeneous/ heterogeneous such as suspended sediments, colloids, organic matter or the introduced solid alkalinization substrates, Moras et al. (2022) reported an $\Omega_{ar}$ threshold of $\sim$5-7 for the observable onset of carbonate formation for the given runtime of the experiment. Potentially, the colloidal structure of $Mg(OH)_2$ precipitates (see Badjatya et al., 2022), typically formed above pH values of $\sim$10.5 as a consequence of alkalinity addition (cf. Eisaman et al., 2023; Haas, 1916; Kapp, 1928; Suitner et al., 2024; Varliero et al., 2024) could serve



the same purpose and lower the threshold for carbonate precipitation. However, the redissolution of the formed
$Mg(OH)_2$ through the mixing and dilution processes, as described by Ringham et al. (2024), may inhibit this effect
and would also allow much higher short-term pH and TA concentrations around an alkalinity injection site when
using liquid stock solutions.
To characterize the transition from a state with negligible shifts in carbonate chemistry towards a phase primarily
driven by carbonate formation (cf. Suitner et al., 2024), a practicable criterion of a 40 µmol kg$^{-1}$ d$^{-1}$ TA-loss was
set to determine the start of the intensified precipitation stage (see Figs. 4 and 5). This criterion was also used to
describe the induction time, which is the period before a measurable onset of secondary carbonate formation can
be detected (Fig. 4). Since the induction time includes a fundamental uncertainty, it does not reflect an intrinsic
property of the treated solution itself and relies on the detection capability of the experimental setup (Söhnel &
Mullin, 1988) and might be chosen differently in future work (see Fig. S9 for varying criteria). While the selected
criterion already depicts relatively high loss rates, it enables detectable changes, distinguishable from measurement
uncertainties or natural variabilities. The overall emerging patterns related to the onset and duration of the APP
nevertheless remained relatively consistent across different tested threshold values.

### 4.3 Predictability of the runaway process

The consistent patterns during the alkalinity loss within all three experimental setups allowed the introduction of
continuous and differentiable functions for each treatment level, enabling further analysis to examine relevant
factors guiding the runaway process. Fuhr et al. (2022) utilized a comparable inverted logistic function to model
the process of secondary carbonate formation during olivine dissolution experiments in seawater. However, the
model was not consistently applied to describe a runaway carbonate precipitation process nor used as a general
predictive model to determine the stability ranges of the added alkalinity in OAE approaches.
The characteristics of the logistic function applied in this study, facilitate the conversion of both empirically
determined and hypothetical parameters, such as induction time, duration of the APP (Fig. 5), or the initial and
final TA levels before and after the runaway process. The applicability of kinetic rate equations, combined with
the ability to quantify the precipitation process, enables a description and prediction of the temporal evolution of
the carbonate formation. This may facilitate the integration of the alkalinity depletion procedure into various
predictive modeling approaches. Although these statements currently apply only under the tested environmental
conditions, they nonetheless suggest the general capability to assess a framework for guiding time and TA level
ranges in OAE approaches. Since the logistic model is based on experimental data from bottle experiments,
processes such as the removal of surface area due to the sinking of precipitated carbonate particles were not
accounted for - see section 4.6 for an approach to address this topic.
Under specified temperature and salinity conditions, as well as predefined TA/DIC levels after OAE treatment and
an estimated final $\Omega_{ar}$ after the precipitation process stopped (typically ~1.5-5.0, see Fuhr et al., 2022; Hartmann
et al., 2023; Moras et al., 2022; Pan et al., 2021; Suitner et al., 2024), the resulting total TA-loss can be computed.
This calculation follows the condition that the alkalinity loss reflects the ideal 2:1 TA:DIC ratio during carbonate
mineral precipitation in seawater (Zeebe & Wolf-Gladrow, 2001). Given these assumptions, upper and lower limits
of the logistic function (coefficients **(a)** and **(d)**, Eq. (1)) can be determined. To characterize measures such as
induction time (coefficient **(b)**) and the duration of the APP (coefficient **(c)**), it is necessary to acquire empirical
data that account for the specific conditions of the deployment area.These data could either be provided by actual
experiments or model predictions, based on a comprehensive database which accounts for broad ranges of TA,





DIC, temperature, salinity, and practical available surface area, as well as inhibitory factors or potential effects of
biota. To validate the predicted precipitation behavior, additional gradient experiments need to be conducted to
better understand the geochemical reaction pathways.

**4.4 Empirical rate equations using $\Omega_{ar}$ and particle surface area during APP**

After passing the induction time to start the detectable carbonate formation process by (pseudo-/)homogeneous
precipitation and overcoming the delaying inhibition effects (Marion et al., 2009; Morse & He, 1993; Schulz et
al., 2023), the triggered heterogeneous precipitation can be described by basic empirical rate equations (Fig. 7, S4
and S5). These equations demonstrate the fundamental role of $\Omega_{ar}$ as a guiding factor for the precipitation process.
The kinetics of carbonate formation remained relatively consistent across all treatment levels within each
experimental approach (see Tabs. 3, S1 and S2). The observed consistent correlations between saturation states
and surface area normalized precipitation rates indicate that the runaway carbonate formation processes during the
present incubations followed the known kinetics of heterogeneous carbonate formation in seawater (cf. Morse et
al., 2007; Zhong & Mucci, 1989).
Fig. 9 illustrates the role of $\Omega_{ar}$ saturation states and generated surface area in guiding the alkalinity loss rates
during the precipitation process. The black graph represents the curve fit of TA-loss rates of the unfiltered neq
$\Delta TA_{2000}$ approach; the experimentally determined rates are indicated by black triangles (cf. Fig. 3c). Assuming
that the entire lost alkalinity was transformed into aragonite precipitates with a surface area of 2.283 m² g$^{-1}$ (see
chapter 2.3), the total generated particle surface area (PSA) could be determined (red dotted graph). The overall
expected TA-loss rate per m² (brown dashed graph) was obtained by utilizing the empirical logistic curve fit for
the temporal evolution of $\Omega_{ar}$ (Fig. 3b), normalizing it to 1 m² surface area, and inserting it into the rate equation
(Eq. (4)). Given that the system initially exhibits a negligible degree of PSA, the relatively high precipitation
potential by the $\Omega_{ar}$ saturation state does not result in a measurable TA-loss rate. Following the presented concept,
the consistently high $\Omega_{ar}$ values led to a continuous (pseudo-/)homogeneous precipitation during the induction
time, thus causing a rise in PSA until the system shifts to heterogeneous precipitation, and ultimately resulting in
a detectable exponential runaway process. The interplay of precipitation potential by $\Omega_{ar}$ and the practical available
surface area could therefore be determined as the primary factors guiding the actual observed TA-loss rates.
Within the uncertainties of the applied calculation steps and methods, the practical TA-loss rate could simply be
described as the product of these two factors (also see Fig. S8 for other treatments). For comparison, the blue data
points in Fig. 9 represent the calculated theoretical loss rates at each sampling day, by inserting the experimentally
determined $\Omega_{ar}$ and PSA values into the related empirical rate equation for $\Delta TA_{2000}$ (see Tab. 3). As this equation
does not account for any inhibitory factors, the resulting rates exhibit a slight positive bias compared to the
observed values.

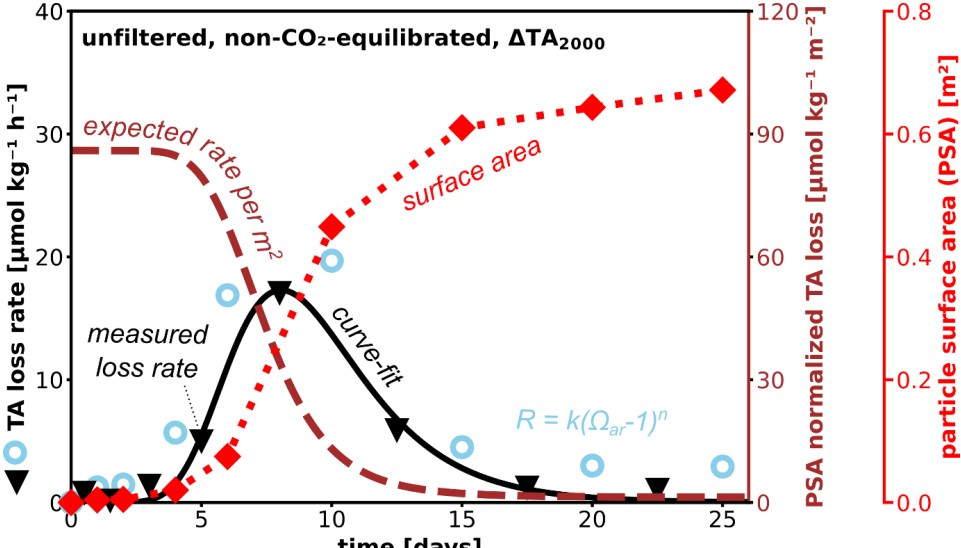

Figure 9: Conceptual figure, illustrating the interplay of $\Omega_{ar}$ and particle surface area guiding the TA-loss rate evolution (dashed, dark red). After TA injection high $\Omega_{ar}$ values provide a high potential for the formation of carbonates by heterogeneous precipitation. In the absence of existing particle surface area (red dotted), (pseudo-/)homogenous precipitation would dominate the period until the start of the APP and the resulting runaway precipitation process. Observed TA-loss rates (black triangles) are therefore a combination of the available practical surface area and the precipitation potential by $\Omega_{ar}$ oversaturation. While the potential to precipitate carbonates decreases with progressive precipitation, additional surface area is generated. The symmetry of the TA-loss rate can be mathematically described with good approximation using only these two factors. Understanding how long particles remain in a critical zone to maintain a full-grown runaway process is therefore relevant for future considerations. Hollow light-blue markers provide the output of the related empirical rate equations for each sampling day. The shown TA and loss rate data are taken from the empirical data sets for the neq unfiltered $\Delta TA_{2000}$ approach, see Fig. S4 for other treatments levels.


## 4.5 Could a runaway process be triggered in an open world context?

Mixing with untreated water around an injection point may lead to an efficient dilution below non-critical alkalinity
levels within seconds to minutes. Such a process would effectively prevent alkalinity leakage, as described in this
study, which assumes that the formed particles act as catalysts for future precipitation. This is supported by findings
from a mesocosm experiment and corresponding side experiments, where the presence of additional suspended
aragonite particles accelerated the alkalinity loss (Paul et al., 2024). In contrast, fresh seawater enhanced to the
same TA-levels did not show any alkalinity loss within 10 days in their experiments. These observations indicate
that free-floating particles in the water column can accelerate heterogeneous precipitation in a runaway style.
Precipitation events can be triggered naturally without additional treatment, especially for locations with already
relatively high $\Omega_{ar}$ background levels, for example, due to high evaporation rates (Bialik et al., 2022) or high
degrees of (re)suspended sediments present on carbonate platforms (Broecker & Takahashi, 1966; Bustos-Serrano
et al., 2009; Morse et al., 2003), or close to river mouths (Wurgaft et al., 2016, 2021) providing additional PSA to
catalyze precipitation events. Under inappropriate alkalinity deployment circumstances, secondary mineral
formation might be triggered locally around injection sites, within short timescales. Moras et al. (2022) suggested



that visible APP starts around $\Omega_{ar}$ of 5, which translates for the mostly particle-free water of the Raunefjorden into
ΔTA ~245 µmol kg$^{-1}$ applying a neq OAE approach and ΔTA ~580 µmol kg$^{-1}$ for an eq OAE approach. The
induction time before the APP begins can be estimated using Equation (5), based on the specified TA-loss criterion
of 40 µmol kg$^{-1}$ d$^{-1}$. For these configurations, the projected induction times would be 1074 days and 143 days,
respectively. However, the predicted induction times lie far outside the calibration ranges specified in this study
and may therefore be inaccurate. Nevertheless, since these projected APP induction times are also within the
suggested residence times of treated water in the upper ocean layers, it is necessary to conduct studies lasting at
least for the projected timespans, depending on the local environmental conditions.
Significantly shorter induction times were identified for subtropical conditions (Temp. ~23°C, Sal. ~36 psu, TA
~2400 µmol kg$^{-1}$). Hartmann et al. (2023) described an onset of the precipitation after just 4 days for a 50µm
filtered neq incubation with initial values of 1050 µmol kg$^{-1}$ for ΔTA and ~15 for $\Omega_{ar}$. Within the same setting,
Paul et al. (2024) observed aragonite formation for a $CO_2$ equilibrated setup with ΔTA ~2300 µmol kg$^{-1}$ and $\Omega_{ar}$
9.74 ± 0.15 in mesocosms after 21 days.

**4.6 Consequence of sinking particles removing surface area for carbonate formation**
Because the TA-loss rate is proportional to the surface area of particles (Eq. (4)), removal of particles due to
sinking processes or dilution with untreated water would result in slower precipitation rates. Small, formed
particles may remain in the upper layer for several months (Fig. 8), while medium-sized particles may leave the
treated water within a couple of days, depending on temperature and salinity conditions (Figs. 8 and S7). Particles
larger than 15 µm are expected to sink within one day under the environmental conditions of the Raunefjorden. If
those particles were removed by sinking while they were still growing, it can be estimated that approximately 30-
40% of the available surface area would be removed from the upper 10 m of the water column within one day
(also see SI). This would decrease the precipitation rate accordingly as surface area and formation rates are linearly
proportional. Potential aggregation would increase the sinking speed and was not considered in this model
calculation but may be relevant in other settings. In general, the abundance and sinking of particles need to be
addressed if the stability or loss of is to be assessed with a high level of confidence.
Efficient dilution of the treated water parcels could therefore significantly reduce ongoing precipitation, especially
if the onset of the APP is initiated within the first few seconds. For example, this could be the case in the wake of
a ship, in OAE applications utilizing existing marine traffic to distribute alkalinity throughout the world's oceans
(Caserini et al., 2021). However, particle-based alkalinization approaches would nevertheless temporarily
introduce additional surface area until its complete dissolution, and may cause the shift into the APP (Hartmann
et al., 2023).



**5 Conclusion**

Alkalinity leakage due to oversaturation sets a limit to the efficiency of OAE approaches. So far, the drivers of the process could not be quantified, preventing the implementation of TA-loss terms in applicability assessments for OAE. An induced runaway process follows predictable patterns that can be modeled using available surface area and aragonite oversaturation, identified as the main factors for the given environmental settings.

However, it is expected that parameterizations will systematically change along temperature and salinity gradients, as well as with naturally occurring variations in particle abundance and quality. The determination of their impact was not within the scope of this work, instead this study aimed to provide a framework for how such needed parameterization can be achieved. Achieving a predictability of the induced TA-loss on a global scale would allow the identification of suitable locations for OAE or optimizing applications. Therefore, further research across salinity and temperature gradients would also enhance the predictive capabilities of ocean models. Runaway TA-loss processes, as described in this study, would be significantly altered under natural conditions by dilution and particle export processes. If sinking of particles and dilution with untreated water are considered, the limitations of laboratory bottle experiments become evident. Nevertheless, they contribute valuable parameterizations for model development. Field experiments are necessary to evaluate the validity of the presented theoretical model framework with respect to dilution and particle sinking processes.



**Data availability**

All datasets will be made available at the time of publication.

**Author contributions**

The idea for this work was conceived by NS, with contributions by JH and SV. NS, SV and PS performed the surface area- and sinking velocity/density measurements. NS interpreted the data with help from all co-authors. NS and JH wrote the text with contributions from all co-authors.

**Acknowledgements**

Peggy Bartsch (UHH), Carl Lim (UHH) and Julieta Schneider (GEOMAR) are thanked for supporting the preparation and execution of the experiments.

**Financial support**

This research has been supported by the German Federal Ministry of Education and Research through the CDRmare projects RETAKE-1: grant no. 03F0895F and RETAKE-2: grant no. 03F0965F; Horizon 2020 (OceanNETs; grant no. 869357); the Deutsche Forschungsgemeinschaft (grant no. 390683824), under Germany's Excellence Strategy (EXC 2037, "CLICCS"; grant no. 390683824) contribution to the Center for Earth System Research and Sustainability (CEN) of Universität Hamburg, as well as the Ocean Alkalinity Enhancement (OAE) R&D Program funded by the Carbon to Sea Initiative.

**Competing interests**

JHA is consulting the Planeteers GmbH. The contact authors have declared that all other authors have no competing interests.





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
