# Peer review of "Surface area and $\Omega$ -aragonite oversaturation as controls of the runaway precipitation process in ocean alkalinity enhancement 2"

_EGUsphere, 2025_

## Referee Comment (RC2)

This study investigates the mechanisms that control runaway calcium carbonate ($CaCO_3$) precipitation during ocean alkalinity enhancement (OAE). The researchers use previously published data to construct a process-based model for estimating total alkalinity (TA) losses due to runaway precipitation. The manuscript is well presented and makes significant contributions to our understanding of runaway calcium carbonate precipitation during OAE deployments.

General comments

The manuscript largely builds upon previously published data from Hartmann et al. (2023) and Suitner et al. (2024). This manuscript does build upon the earlier work; however, the authors frequently direct the reader to these articles throughout the manuscript. In several instances I believe it would be beneficial to simply provide the information within this manuscript rather than directing the reader to another article, while in other circumstances I do not feel it necessary to continuously cite these articles. Lines 160-161 direct the reader to the supplementary information of Suitner et al. (2024) while this information if important should be included in this manuscript or its supplementary information. Also, lines 252-259 cite the articles 5 times, in particular line 253 refers the reader to Suitner et al. (2024) for images of the aragonite precipitates which are also depicted in the current manuscript in figure 8.

To my knowledge increases in total alkalinity of 1600 – 2800 µmol/kg above current levels is not realistic in real world settings. The authors state themselves that the standard experimental setup (which used relatively high delta TA values) did not provide enough precipitates. This raises questions about the applicability of such extreme perturbation studies to real-world scenarios. Understanding such processes are undoubtedly important, however their relevance to actual OAE seems somewhat limited. I would welcome further discussion around this point expanding upon section 4.5. Additionally, the authors utilise mesocosm and small-scale bottle experiments, while the limitations of such experiments are well understood a short discussion of the limitations of such datasets would be beneficial.

In contrast to the discussion surrounding omega and precipitation under high alkalinity values I feel the manuscript could discuss the influence of suspended particles further. This factor appears to be much more likely to result in runaway precipitation than the intentional increase in omega aragonite above 20, particularly as researchers and companies look to rivers to transport alkalinity to the ocean.

Specific comments

Comment 1

Figure 4. This is an extremely busy figure, and I would recommend removing some of the text from within the actual figure and placing it outside. For example, the APP explanation, breaks up the plot to a point where it seems as if there are two distinct plots. A distinct legend would likely be beneficial, and I recommend the authors consider this as well as simplifying/removing some of the text within the plot.

It is unclear as to why the triangles for the TA1200 treatment are hollow and smaller in comparison to the other two treatment levels utilise filled triangles.

The secondary x axis is difficult to interpret as it appears to show start and end deltaTA values for CO2 equilibrated but only start values for the unfiltered and end values for filtered non-co2 equilibrated measurements. Proper axis labelling here would be beneficial or removal of this secondary x axis.

Comment 2

Both figures 4 and 5 depict TA on the x axis and time on the y axis, however all other figures in the manuscript appear to utilise the opposite axis labelling. This is confusing. I strongly recommend the authors use consistent labelling of the axis throughout the manuscript.

Comment 3

Lines 222 – 224. Suggest that the initiation of APP may be estimated via initial TA or omega aragonite values. However, figure 6 illustrates this to also be dependent on the approach and assumingly particle density. I appreciate this is highlighted in the figure caption but believe this should be explicitly stated in the text.

Comment 4

Line 299: Might help prevent

Comment 5

Line 390: it is unclear what cf. stands for and no black triangles are present in fig 3c

**Comment 6**

Line 389: description of the line/fits in figure 9 throughout this section are misleading please change wording from "graph" to "line/fit" throughout.

**Comment 7**

I appreciate section 4.5 and the discussion surrounding the context of this experiment in a real-world setting. However, lines 427 – 429 state "Nevertheless, since these projected APP induction times are also within the suggested residence times of treated water in the upper ocean layers, it is necessary to conduct studies lasting at least for the projected timespans, depending on the local environmental conditions". I question whether a perturbed water parcel would remain in its perturbed state given the physical processes occurring in the surface ocean. Irrespective of its residence time, if the water parcel becomes diluted the omega values which the authors state as a controlling factor for determining precipitation would be significantly lower rendering any further precipitation highly unlikely.

**Comment 8**

I question the use of stokes law which as stated by the authors is used for solid spherical particles. Given the SEM images provided by the authors the particles appear to have significant cavities likely increasing the SA/V ratio and thus significantly influencing the sinking velocity. It is also unclear whether measured or calculated sinking velocities are used.

**Comment 9**

I question the relevance of using 10 m as the upper water layer given mixed layer depths often range from 10 – 100m. Understandably it is important that the perturbed water remains in contact with the overlying atmosphere for CO2 uptake to occur. However, if one is to consider the removal of particles from a system, they must consider it in the context of the MLD. Any particle flux above the MLD at the current point in time is not equivalent to the removal of the particle from the discussed water layer as it is entirely possible that the particle may re-enter the 10 m layer due to various mixing processes.

Further to this I question if the authors considered two important aspects of particle fluxes 1) attenuation with depth and 2) variable mixed layer depth and reinjection of particles into the surface layer. Understandably the particles described here are abiotic but a comment on the potential attenuation of these particles would be appreciated. Following this, have the authors considered reinjection of the particles via mixing

processes? Although unlikely for the fast-sinking fraction I question if this may enable a delayed CO2 uptake or alter the OAE efficiency.

Comment 10

I appreciate the authors thoughts surrounding the transport of particles and their subsequent effect on runaway precipitation. However, given the context of this study I believe an estimation of the effect of particle export on OAE efficiency would be beneficial. Especially considering that the sinking velocities and abundance of particles have been calculated. I also again query how the authors differentiate between small particles capable of staying in the upper layer for months and those which sink "while still growing". Could it be more realistic to assume that most particles continue to grow until they aggregate or reach a sinking threshold? And if so, how would this translate to OAE efficiency? Particularly given that the removal of growing or fully grown particles would likely begin to dissolve as they descend the vertical water column and have differing impacts on the alkalinity export.

Comment 11

Supplementary figure S1 could benefit from some slight adjustments so that the brightness of each image is similar. S1.B is very dark making it difficult to see any important details such as the branching shown in figure S1 A and B.

Comment 14

Image quality of the supplementary figures S2, S3, S7, S8, S9 and S10 is poor and should be improved prior to publication.

Comment 15

Figure S8 describes plots as the interplay between omega aragonite and surface area of particles. However, I would argue that it is the omega aragonite controlling the size of the precipitated particles and thus surface area.

Comment 16

Figure S10. I appreciate the lengthy explanation by the authors here however they fail to link such particle transformations back to the primary purpose of OAE, namely CDR. It is important to understand how such particle transport mechanisms would act upon

the overall efficiency of the OAE deployment not just the particle surface area. To expand, an export of particles due to caco3 precipitation is still a loss of alkalinity and thus a reduction in efficiency. What would be interesting is to understand at what point this reduction in efficiency is beneficial if at all? Is there a point at which such extreme perturbations would be beneficial over smaller or medium sized perturbations. Otherwise, such a discussion may have limited relevance unless connected directly to implications for CDR efficiency.

---

## Author Comment (AC1)

**Reply to comments on EGUSPHERE-2025-381**

RC2: 'Comment on egusphere-2025-381', Anonymous Referee #2, 25th June 2025

Citation: (https://doi.org/10.5194/egusphere-2025-381-RC2)

This study investigates the mechanisms that control runaway calcium carbonate (CaCO3) precipitation during ocean alkalinity enhancement (OAE). The researchers use previously published data to construct a process-based model for estimating total alkalinity (TA) losses due to runaway precipitation. The manuscript is well presented and makes significant contributions to our understanding of runaway calcium carbonate precipitation during OAE deployments.

**General comments**

The manuscript largely builds upon previously published data from Hartmann et al. (2023) and Suitner et al. (2024). This manuscript does build upon the earlier work; however, the authors frequently direct the reader to these articles throughout the manuscript. In several instances I believe it would be beneficial to simply provide the information within this manuscript rather than directing the reader to another article, while in other circumstances I do not feel it necessary to continuously cite these articles. Lines 160-161 direct the reader to the supplementary information of Suitner et al. (2024) while this information if important should be included in this manuscript or its supplementary information. Also, lines 252-259 cite the articles 5 times, in particular line 253 refers the reader to Suitner et al. (2024) for images of the aragonite precipitates which are also depicted in the current manuscript in figure 8.

To my knowledge increases in total alkalinity of  $1600-2800~\mu mol/kg$  above current levels is not realistic in real world settings. The authors state themselves that the standard experimental setup (which used relatively high delta TA values) did not provide enough precipitates. This raises questions about the applicability of such extreme perturbation studies to real-world scenarios.

Understanding such processes are undoubtedly important, however their relevance to actual OAE seems somewhat limited. I would welcome further discussion around this point expanding upon section 4.5.

Additionally, the authors utilise mesocosm and small-scale bottle experiments, while the limitations of such experiments are well understood a short discussion of the limitations of such datasets would be beneficial.

We thank the reviewer for the timely and valuable comments. These insights have greatly contributed to improve this work, particularly regarding particle surface and sinking characteristics, and has helped us to better reflect the model considerations provided. Below, we provide a point-by-point response to each of your comments.

As requested, references to Hartmann et al. (2023) and Suitner et al. (2024) will be reduced, with essential information from these works incorporated directly into this manuscript.

We would like to clarify that no new experiments were conducted for this study, except for a small-scale precipitation setup used to generate sufficient precipitate material for BET surface area measurements. Additionally, sinking velocity measurements using the precipitates from the campaigns in Bergen and Gran Canaria were published in this work.

The main focus of this work was to compile published data to develop an alkalinity loss model, which is controlled by two parameters, the  $\Omega_{aragonite}$  and the surface area of abundant particles, represented by precipitates formed during the experiment. The main experiment was conducted in 2022, and at that time, little was known about the full parameter space within which alkalinity is stable. The same applies to the time period or speed of a runaway precipitation process once critical values have been surpassed. To our knowledge, a comprehensive description of such a parameter space for stable

alkalinity enhancement, post-treatment time spans to trigger runaway processes, and rates of TA-loss have never been presented in such detail.

It is probable that if OAE were deployed, the target alkalinity levels in the regional or local water body as a whole would be lower than the high values used in this work. However, we would like to point out that at the release points of alkalinity, in whatever form chosen, TA is likely to be enhanced above the actual target level. For example, when particles are used for alkalinization, it is very likely that carbonate particles precipitate (see Moras et al., 2022, Hartmann et al., 2023). As shown in this work, the amount of added alkalinity and surface area influence the initiation of the runaway processes and the related TA loss rate. In addition, to calibrate the alkalinity loss model based on  $\Omega_{\text{aragonite}}$  and the abundant surface area from particles, it is necessary to consider extreme cases to achieve accurate calibration results. Specifically, the high alkalinity levels used in this study were useful to understand the shape and continuous evolution of the loss function, as well as identifying which functional terms should be used. To clarify the reasoning behind the chosen level of perturbation, additional information will be provided in the manuscript.

There is indeed considerable debate within the scientific community about the relevance of studies that test very high levels of alkalinity. However, we argue that exploring the full range of possible parameter spaces is necessary for optimal best practice predictions of potential TA loss processes. Waste water management facilities, for example, often release highly alkalized waters regularly, in a range of 4000-6000  $\mu$ mol kg-1. Therefore, the chosen  $\Delta$ TA are not unrealistic for the near field space around the injection site. The complementing additional BET- and FlowCam measurements were needed to provide the necessary input to calibrate the empirical rate equation and the sinking velocity.

In contrast to the discussion surrounding omega and precipitation under high alkalinity values I feel the manuscript could discuss the influence of suspended particles further. This factor appears to be much more likely to result in runaway precipitation than the intentional increase in omega aragonite above 20, particularly as researchers and companies look to rivers to transport alkalinity to the ocean.

Following the reviewer's suggestion, we will discuss this further in the new version of the paper by addressing these points. Specifically, we will describe more clearly the importance of surface area as a controlling factor in precipitation events. That is also one of the reasons why we have been conducting the additional BET surface area measurement and the size and shape description of hundreds of precipitated particles. We are aware of the discussion around river alkalinity enhancement. Just recently, we published a work focusing on the TA-loss after river alkalinity enhancement (see Tian et al., 2025, ERL), addressing these points.

**Specific comments**

**Comment 1**

Figure 4. This is an extremely busy figure, and I would recommend removing some of the text from within the actual figure and placing it outside. For example, the APP explanation, breaks up the plot to a point where it seems as if there are two distinct plots. A distinct legend would likely be beneficial, and I recommend the authors consider this as well as simplifying/removing some of the text within the plot.

The APP explanation and legend will be placed outside of the diagram space to simplify the visual impression.

It is unclear as to why the triangles for the TA1200 treatment are hollow and smaller in comparison to the other two treatment levels utilise filled triangles.

The triangles will be standardized for consistency. Hollow markers were chosen to enhance readability when overlapping. Thank you very much for pointing out this issue.

The secondary x axis is difficult to interpret as it appears to show start and end deltaTA values for CO2 equilibrated but only start values for the unfiltered and end values for filtered non-co2 equilibrated measurements. Proper axis labelling here would be beneficial or removal of this secondary x axis.

 $\Delta$ TA 1200 and 1400 in the non-CO2-equilibrated and  $\Delta$ TA 4400 in the CO2-equilibrated approaches are only thematized in this diagram. The purpose of these additional markers was to provide a simple visual orientation to clarify the added alkalinity compared to the baseline. We would like to keep this information, as some readers may find it beneficial for orientation.

**Comment 2**

Both figures 4 and 5 depict TA on the x axis and time on the y axis, however all other figures in the manuscript appear to utilise the opposite axis labelling. This is confusing. I strongly recommend the authors use consistent labelling of the axis throughout the manuscript.

The diagrams follow the rationale of plotting the variable on the x-axis and the output value on the y-axis. We acknowledge that using time as an output could be misleading. Normally, the depending variable of a functional relationship is plotted on the y-axis. The intention was to depict the induction time (Fig. 4 and Tab. 2) and the APP timespan (Fig. 5) in dependence on the initial TA level, meaning e.g., that induction time = f(TA)). Therefore, we would like to recommend maintaining the current version.

**Comment 3**

Lines 222 – 224. Suggest that the initiation of APP may be estimated via initial TA or omega aragonite values. However, figure 6 illustrates this to also be dependent on the approach and assumingly particle density. I appreciate this is highlighted in the figure caption but believe this should be explicitly stated in the text.

We will add the information that APP is also a function of surface area into the main text in section 3.4: "[...] these functions could therefore assess the initiation of the APP for specific initial TA or  $\Omega_{\text{aragonite}}$  levels based on a given starting particle surface area [...]".

**Comment 4**

Line 299: Might help prevent

It will be corrected, thanks a lot for noticing.

**Comment 5**

Line 390: it is unclear what cf. stands for and no black triangles are present in fig 3c

"cf." is used as an abbreviation for "compare" to refer to a reference or figure throughout the text. In this case, the reference "(cf. Fig. 3)" is ambiguous. The referred black triangles can be found in Fig. 9. "(cf. Fig. 3)" will be changed to "(also see Fig. 3)".

**Comment 6**

Line 389: description of the line/fits in figure 9 throughout this section are change wording from "graph" to "line/fit" throughout.

Wording will be standardized.

**Comment 7**

I appreciate section 4.5 and the discussion surrounding the context of this experiment in a real-world setting. However, lines 427 – 429 state "Nevertheless, since these projected APP induction times are also within the suggested residence times of treated water in the upper ocean layers, it is necessary to conduct studies lasting at least for the projected timespans, depending on the local environmental conditions". I question whether a perturbed water parcel would remain in its perturbed state given the physical processes occurring in the surface ocean. Irrespective of its residence time, if the water parcel becomes diluted the omega values which the authors state as a for determining precipitation would be significantly lower rendering any further precipitation highly unlikely.

We thank the reviewer for this comment. We agree that for a single point source in an average open ocean setting, a persistent runaway process might be unlikely. However, so far, it is unclear how and where alkalinity addition would take place in practice. Large-scale arrays could potentially be employed to scale up OAE to an industrial level and it would be possible that inappropriate treatment could result in an initial seeding with small-sized precipitates. Assuming these particles are not dragged down by physical processes, they would lower the precipitation threshold to around  $\Omega_{aragonite}$  5 in the near-field, thereby reducing the overall efficiency. Similar patterns may also apply in coastal regions, bays, and lagoons, as well as in slurries or stock solutions that utilize seawater as a feedstock for alkalinization. Further discussions on the consequence of particles and surface area removal is provided in the responses to Comment 9 and 10. The goal of this work is to provide a tool for estimating TA loss through a process-based formulation, employing established methods from aquatic geochemistry, and to assess its plausibility. Following the analysis of patterns in this study, which identified  $\Omega_{aragonite}$  and surface area as the main drivers of potential carbonate precipitation, the next step to develop a more comprehensive model would involve assessing TA loss in relation to particle abundance, whether from natural background suspension loads or introduced solid phases for alkalinity enhancement. The presented functions are independent of initial conditions and suitable for application to scenarios where manipulated water is mixed with untreated water, enabling their implementation in future model considerations.

Finally, we agree with Referee #2's comments and suggest that local test sites should be chosen in an open-world setting to further explore and understand related processes and whether the identified functions described in this work are actually applicable.

**Comment 8**

I question the use of stokes law which as stated by the authors is used for solid spherical particles. Given the SEM images provided by the authors the particles appear to have significant cavities likely increasing the SA/V ratio and thus significantly influencing the sinking velocity. It is also unclear whether measured or calculated sinking velocities are used.

We thank the reviewer for this comment. To calculate and standardize the sinking velocities, the common practice of equivalent spherical diameters (ESD) is used, following the methods described in (Bach et al., 2012), also used for the FlowCam measurements in this work. As stated in the text, such simplification might come with certain difficulties: "However, most particles are not spherical and contain numerous cavities within their structure, which likely contributes to an underestimation of particle densities." (L276-278) [Reworked as a request of Referee #1 to: "However, most particles are non-spherical and contain numerous internal cavities within their structure (see Fig. S1), and their densities are therefore expected to be lower than those of pure aragonite"], but we believe that it allows a practical way to operate within reasonable accuracy. Taking into account the characteristics of every single shape or even just categories of shapes would result in a level of complexity which is out of the scope of this work. Nevertheless, considering the cylindrical or dumbbell shaped particles with rough uneven surfaces with irregular cavities would potentially yield values closer to reality. The sinking velocity measurements were used to determine the density of the precipitates. The outcome (~2.358 g/cm³) was then used to calculate the sinking velocities for particles following the ESD practice.

**Comment 9**

I question the relevance of using 10 m as the upper water layer given mixed layer depths often range from 10 – 100m. Understandably it is important that the perturbed water remains in contact with the overlying atmosphere for CO2 uptake to occur. However, if one is to consider the removal of particles from a system, they must consider it in the context of the MLD. Any particle flux above the MLD at the current point in time is not equivalent to the removal of the particle from the discussed water layer as it is entirely possible that the particle may re-enter the 10 m layer due to various mixing processes.

Thank you very much. We would like to clarify that it was not stated that a mixed layer depth of 10 m was assumed. Indeed, 10 m was used as an estimate for the thickness of an alkalized plume. The mixed layer depth was set to: "[...] mixed layer depth (assumed to be 200m), [...]" (Caption Fig. 8, L292). In addition, the sinking/growth model presented in this study, is a first-order model estimate; and detailed sinking and/or fluid dynamics are not in focus of this work. Future model assumptions should be refined if such processes are relevant to determine the efficiency of an OAE application scenario.

Further to this I question if the authors considered two important aspects of particle fluxes 1) attenuation with depth and 2) variable mixed layer depth and reinjection of particles into the surface layer. Understandably the particles described here are abiotic but a comment on the potential attenuation of these particles would be appreciated.

These are important points to address in future work, given the complexity modelers already face when predicting the distribution of alkalinity alone. Future collaboration with modelers is anticipated in order to better constrain the particle fluxes and their distribution within the water column over time. We acknowledge that accurately capturing particle reactivity as growth seeds, along with the depth-dependent evolution of particle size distribution, growth during sinking, and varying saturation

levels, represents a considerable modeling challenge. Given the length and complexity of this work, we would recommend to wait for further adapted sinking and particle redistribution models, which might be coupled with the here presented TA-loss model in the future.

The discussion on this topic can be found from line 440 to 447: "Particles larger than 15  $\mu$ m are expected to sink within one day under the environmental conditions of the Raunefjorden. If those particles were removed by sinking while they were still growing, it can be estimated that approximately 30- 40% of the available surface area would be removed from the upper 10 m of the water column within one day (also see SI). This would decrease the precipitation rate accordingly as surface area and formation rates are linearly proportional. Potential aggregation would increase the sinking speed and was not considered in this model calculation, but may be relevant in other settings. In general, the abundance and sinking of particles need to be addressed if the stability or loss is to be assessed with a high level of confidence."

Following this, have the authors considered reinjection of the particles via mixing processes? Although unlikely for the fast-sinking fraction I question if this may enable a delayed CO2 uptake or alter the OAE efficiency.

Aragonite particles formed in the water column typically do not dissolve in standard seawater, with the possible exception of specific anoxic conditions. Therefore, the resurfacing of the particles will not lead to a larger  $CO_2$ -uptake of seawater, as they would not increase the  $CO_2$ -uptake potential. However, if particles sink below the carbonate compensation depth and dissolve, they may contribute to additional "redissolved"  $CO_2$  uptake potential in the water. Since deep water is often supersaturated with respect to  $CO_2$ , we assume that the outgassing potential of upwelling deep water might actually decrease. However, the modeling and evaluation of such processes are beyond the scope of this study, and we refer to published and ongoing research by Earth system modelers addressing these specific questions.

**Comment 10**

I appreciate the authors thoughts surrounding the transport of particles and their subsequent effect on runaway precipitation. However, given the context of this study I believe an estimation of the effect of particle export on OAE efficiency would be beneficial. Especially considering that the sinking velocities and abundance of particles have been calculated. I also again query how the authors differentiate between small particles capable of staying in the upper layer for months and those which sink "while still growing". Could it be more realistic to assume that most particles continue to grow until they aggregate or reach a sinking threshold? And if so, how would this translate to OAE efficiency? Particularly given that the removal of growing or fully grown particles would likely begin to dissolve as they descend the vertical water column and have differing impacts on the alkalinity export.

In the main text, we provide a first-order estimation for a simple scenario to assess whether the sinking process is relevant, and also question ourselves on how to better represent these processes. We reached out to modelers in Canada, the UK, and Germany to explore the most robust way to represent this. The experts estimate that developing a model capable of reproducing the features mentioned by Referee #2 would take years. We consider these points to be relevant and suggest waiting until the processes can be modeled in more detail.

In the presented first-order model, all particles are considered to grow over time. Aggregation is not considered in the calculation: "Potential aggregation would increase the sinking speed and was not considered in this model [...]" (L445).

**Comment 11**

Supplementary figure S1 could benefit from some slight adjustments so that the brightness of each image is similar. S1.B is very dark making it difficult to see any important details such as the branching shown in figure S1 A and B.

We will adjust the brightness of each image in Fig. S1.

**Comment 14**

Image quality of the supplementary figures S2, S3, S7, S8, S9 and S10 is poor and should be improved prior to publication.

Yes, we agree, and vector graphics have already been provided to the journal.

**Comment 15**

Figure S8 describes plots as the interplay between omega aragonite and surface area of particles. However, I would argue that it is the omega aragonite controlling the size of the precipitated particles and thus surface area.

As the  $\Omega_{aragonite}$  evolution is predominantly driven by the  $[CO_3^{2-}]$  concentration, it is mainly dependent on the TA and pH evolution and their effect on the carbonate system. The same accounts for the mass and therefore the surface area of precipitated particles. So, both parameters are related and their evolution roots from the same process.

**Comment 16**

Figure S10. I appreciate the lengthy explanation by the authors here however they fail to link such particle transformations back to the primary purpose of OAE, namely CDR. It is important to understand how such particle transport mechanisms would act upon the overall efficiency of the OAE deployment not just the particle surface area. To expand, an export of particles due to caco3 precipitation is still a loss of alkalinity and thus a reduction in efficiency. What would be interesting is to understand at what point this reduction in efficiency is beneficial if at all? Is there a point at which such extreme perturbations would be beneficial over smaller or medium sized perturbations. Otherwise, such a discussion may have limited relevance unless connected directly to implications for CDR efficiency.

We agree that an assessment of the effect of particle generation and sinking on the overall efficiency would be of relevance. The first order and 1-D character of the presented sinking model might unfortunately, not be an appropriate approach to realize this in a relevant manner. Since this is only a site calculation exploring whether this process could be relevant, we believe it is appropriate to argue that further studies with more complex models are studies on their own. This study wanted to evaluate whether a process-based model for TA-loss rates can be developed and validated. Further investigation and extension of the sinking/efficiency effects might be a bit out of focus for this work, but could be a topic for the future. Thank you very much for pointing it out.

---

## Author Comment (AC2)

**Reply to comments on EGUSPHERE-2025-381**

RC1: 'Comment on egusphere-2025-381', Anonymous Referee #1, 9th June 2025

Citation: (https://doi.org/10.5194/egusphere-2025-381-RC1)

Suitner and colleagues tackle the important question of runaway precipitation as a potential effect of ocean alkalinity enhancement. They use experimental data to derive logistic functions to describe the progression and identify distinct phases of the runaway precipitation process. The aragonite saturation state and available nucleation surface area are identified as key drivers behind the precipitation process. The authors conclude by speculating about the impact of particle removal through sinking on the possible magnitude of runaway precipitation in natural systems.

While the authors present interesting data, I struggle to see how the study advances our understanding of runaway precipitation. My main criticism is that the logistic function presented as a conceptual model is not generalised and anchored to critical environmental factors. The coefficients for the functions are derived from each experimental dataset and are thereby not generalisable (which the authors acknowledge). No attempt is made to describe how the aragonite saturation state and available surface area for precipitation impact the coefficients of the logistic function, despite these factors being identified as key drivers behind the precipitation process in this very study. The equation also does not account for temperature, salinity, or the concentrations of known precipitation inhibitors. As such, the suggested logistic models have no predictive power outside of the specific experimental conditions from which they were extracted. I would encourage the authors to reassess their data and possibly conduct follow-up experiments to mathematically describe critical environmental and chemical factors, which would result in a true conceptual model. That runaway precipitation follows a logistic function was already concluded by Suitner et al. (2024), so I do not see this manuscript as a substantially novel contribution.

Thank you very much for your feedback. We appreciate the level of detail in your comments. You will find a point-by-point response to each comment below.

**Reply to the general comment:**

The initial idea of this work was to refine the analysis of the existing experimental datasets from Suitner et al. (2024), as major aspects, like the rate equations or a thorough description of main drivers of the precipitation process, have not been discussed in detail. Said so, we agree with the reviewer that the next fundamental step would be the implementation of critical environmental factors (e.g. temperature, salinity, sediment load, inhibitors, etc.) into the provided equations. We anticipate that a whole series of experiments aiming to achieve this goal are ongoing or planned within the next years. We agree that a single setup is insufficient to generalize the described functions and patterns. Therefore, more settings with varying environmental factors should be and are currently being investigated. Nevertheless, the existing data also appear to provide sufficient indications for developing generalizable descriptions of overall patterns. L297-298: "For the present study, the compiled concepts allowed the description of the principles guiding the entire runaway process in site-specific condition". The general "shape" and the key drivers of the process have been identified and cross-connections pointed out.

The logistic function is thereby employed as a suitable fit to describe the "shape" of the TA loss/precipitation process by transforming discrete data series into continuous functions. It was not intended to expand the related curve fits into an overarching predictive model. For the present setup, the characterized continuous functions combined with the classical rate equations seem to outline the measured data quite well.

L298-303: "The obtained capability to predict TA-stability ranges, in terms of time and magnitude, might help preventing secondary mineral formation, thereby optimizing the assessments for OAE

application scenarios. Furthermore, the simplicity of the logistic curve fit model, along with the demonstration that the carbonate precipitation follows simple rate law equations "[...], might facilitate the straightforward integration of these fundamental mechanisms into ocean models [...]"

A future potential predictive model might therefore be derived from rate equations, with substituted logistic curve-fits for the evolution of  $\Omega_{ar}$  and/or the particle surface area. Environmental factors would typically modify the rate equation rather than the logistic curve fit of carbonate chemistry parameters.

The methodology is only described very briefly and relies heavily on citations of previous work. Even if a full repetition of methods is not needed, a more extensive description is required in the current manuscript to explain the limits and results of the study.

It was a deliberate decision to cut the method section, as this work is almost entirely theoretical. We acknowledge the reviewer's request, and an additional chapter describing the methodology of the data acquisition studies will be added to improve clarity. We would like to suggest to provide a short overview in the main text, while providing a more detailed version in the supplements as well.

Finally, I encourage the authors to publish the data either as a supplement to the manuscript or as a separate dataset.

All datasets will be made fully accessible during/after the publication process. Either via Biogeosciences or through a data repository like PANGAEA or Zenodo.

**Minor comments:**

TA and alkalinity are used interchangeably throughout the text; be consistent.

For consistency, total alkalinity will be referred to as TA throughout the entire text. Thanks for noticing.

L46-48: I am sure this is not the authors' intention, but to me, the sentence suggests that NETs can be seen as an alternative to emission reductions. Consider rephrasing.

It will be rephrased accordingly:

**[New]:**

"To mitigate climate change and reach net-zero greenhouse gas emissions by the end of the century, negative emission technologies (NETs) are necessary **besides greenhouse gas emission reduction** (UNFCCC, 2015) considering the slow change in the development of the energy infrastructure, lifestyle of humanity, and national goals for economic growth (Fuss et al., 2018; lyer et al., 2015; Sers & Victor, 2018)."

**[Old]:**

"To mitigate climate change and reach net-zero greenhouse gas emissions by the end of the century, negative emission technologies (NETs) are necessary (UNFCCC, 2015) considering the slow change in the development of the energy infrastructure, lifestyle of humanity, and national goals for economic growth (Fuss et al., 2018; lyer et al., 2015; Sers & Victor, 2018)."

L91: Here, it seems like runaway precipitation is a desired phenomenon. Consider rephrasing.

Sentence slightly changed to:

**[New]:**

"To sustain a triggered runaway carbonate formation (Fig. 1), it is necessary to retain the precipitates in the system."

**[Old]:**

"To sustain the observed runaway carbonate formation (Fig. 1), it is essential to retain the precipitates in the system."

Section 2.1: The description of the experimental setup is too brief; it is not enough to refer to Suitner et al. (2024). Please include information about initial TA concentrations and aragonite saturation states, number of replicates, samples collected, analysis methods and uncertainties, etc. The Gran Canaria setup should also be briefly introduced here, since seemingly new results from that experiment are presented in this manuscript.

Thank you very much for the comment. The description of the experimental setup from the Suitner et al. (2024) study will be expanded and the requested information given. Since only the grain size distribution of the precipitates from the Gran Canaria study is included here, the authors suggest limiting the description of the study to a minimum, approximately 2-3 sentences to provide a brief overview. As suggested above, we would also include a more detailed overview of the methods in the supplementary materials.

L131-132: How was the assumed available active mineral surface area obtained? As described below?

The related methodology is described in L134-142

L136: Filtered or unfiltered seawater?

Seawater was 0.2 µm filtered – this information will be added in L136

L139: The BET surface area is a result, and should preferably include an uncertainty as well.

We thank the reviewer for pointing this out. In the new version, the results from multiple measurement runs and the associated uncertainty will be reported.

Section 3.1: The data from treatments that did not experience runaway precipitation are not presented. Please include at least in the supplementary material. Furthermore, please add a table with the coefficients of the logistic functions to the supplementary material.

A table with the coefficients of the logistic functions and plots including all treatment levels, including plots for those without precipitation, will be provided in the supplements as requested. Please note that no curve fitting was performed for treatment levels without precipitation.

L160-161: The removal of outliers needs to be described in detail here, as the entire manuscript is based on curve fitting. I also suggest including removed data points in Fig. 3 as empty symbols.

Text and plots (Fig. 3 as well as S2 and S3, which might be moved to the main text – see next comment) will be adjusted as suggested.

Figure 3: The filtered neq and filtered eq treatments show quite different patterns from the unfiltered neq treatment, and I think those figures should be shown in the main text.

We thank the reviewer for this suggestion; however, after some discussions, we have decided not to include all three plots in the main text, as it would overload the manuscript and not provide additional insights into the overall process. Displaying these datasets would also necessitate presenting all related tables with coefficients and at least some, if not all, associated conceptual figures (such as those in Fig. 9 and S8). Regarding the filtered neq approach, deviations in the patterns originate from various treatment levels that triggered the APP almost immediately. Consequently, the corresponding diagram would appear overloaded with information, but it would show similar behavior in detail. The differing characteristics of a CO2-equilibrated approach naturally result in diverging absolute numbers, but the method for determining the logistic function remains the same. Furthermore, both requested diagrams have already been presented in Suitner et al. (2024), and reproducing them would simply replicate results already available there.

Figure 3A-B: I suggest adding the TA concentration and aragonite saturation state of the initial seawater to the figure as well.

We appreciate the suggestion, it will be implemented as recommended.

L192-194: Please include as a figure in the supplementary material.

It will be provided as requested.

L219: Confusing phrasing, there is a decrease in APP timespan with increasing initial TA.

Thank you very much for noticing. This will be rephrased to:

**[New]:**

"In the eq approach, the APPs showed a continuous decrease as the initial TA addition levels increased, ranging from 5 to 11 days."

**[Old]:**

"APPs in the eq approach showed a continuous increase ranging from 5 to 11 days."

Section 3.4: I find this section somewhat confusing. The coefficients b and c determine the shape of the logistic function, so it is only natural that they correlate well with the induction time and APP timespan (which are determined from the shape of the logistic function).

The idea of this section is to provide simple equations for the conversion of abstract coefficients to relatable parameters. The high correlation is therefore a natural consequence, as pointed out by Referee #1. A sentence will be added in Section 3.4 to clarify this point.

**Section 3.5: The manuscript is generally well written; however, I struggled with this section. Please go over the text again.**

We would like to suggest the following changes in L232-244:

**[New]:**

"Additional insights into the reaction speed and the associated timespan of the APP can be obtained through analysis of empirical rate law equations. As an example, Fig. 7 illustrates the relationship between the logarithm of TA-loss rates normalized to the surface area and the aragonite saturation states for the unfiltered neq approach (see Figs. S4 and S5 for details on the filtered approaches), focusing on treatments that entered the APP. Throughout all experiments the logarithm of the surface area normalized TA-loss rates R correlates with the  $\log(\Omega_{ar}-1)$ , in accordance with similar observations reported in literature (e.g. Morse et al., 2007; Mucci & Morse, 1983; Zhong & Mucci, 1989). The parameters n and k in Eq (2):  $R = k(\Omega_{ar}-1)^n$  were determined for each treatment level in this work, as outlined in Eq (2) to (4) (section 2.3). Here, R represents the surface area normalized TA-loss rate, and k denotes the rate constant.

The values for n and log(k) derived from the linear regressions in the unfiltered neq treatments are provided in Tab. 3 (see Tab. S1 and S2 for filtered experiments). These values demonstrate reasonable consistency in n and log(k) within each of the three separate experiments. Treatment levels influenced by the immediate formation of  $Mg(OH)_2$  as pH approached approximately 10.3 show minor deviations, the remaining treatment levels exhibit reaction orders (n) within a relatively narrow range of 2.45 to 2.73. In comparison, log(k) values ranged between 0.30-1.68, showcasing a higher variability."

**[Old]:**

"Further implications about the reaction speed and the related timespan of the APP can be provided by empirical rate law equations. As an example, Fig. 7 demonstrates the relationship between the logarithm of TA-loss rates normalized by surface area and aragonite saturation states for the unfiltered neq approach (see Fig. S4 and S5 for details on filtered approaches), for treatments that entered the accelerated precipitation phase. Throughout all experiments  $\log(R)$  TA-loss rates correlate with the  $\log(\Omega ar-1)$ , expressing the characteristic relationship for carbonate formation (see Morse et al., 2007; Mucci & Morse, 1983; Zhong & Mucci, 1989). The strong correlation of the linear regressions within each experiment enables the articulation of the empirical rate equations, such as Eq. (2):  $R = k(\Omega_{ar}-1)^n$ . In this equation R represents the surface area normalized TA-loss rate, k the rate constant and n the reaction order. The related values for n and  $\log(k)$  derived from the linear regressions are provided in Tab. 3 (see Tab. S1 and S2 for filtered experiments), showing reasonable consistency in n and  $\log(k)$  values within each of the three separate experiments. While some treatments, showing immediate Mg(OH)2 formation, slightly deviate, the other treatment levels displayed reaction orders (n) within a relatively narrow range of 2.45 to 2.73. In comparison,  $\log(k)$  values ranged between 0.30-1.68, showcasing a higher variability."

**Section 3.6: Here, it would be interesting to compare n and k between treatments, so I think it would be relevant to show data for filtered neq and filtered eq as additional panels in Figure 7 and Table 3.**

Thank you very much for the suggestion. So far, the data for filtered setups are just presented in the supplements, and the main body of the text is consequently only showing the unfiltered treatments, assumed to be the closest to natural seawater conditions. As already stated above, we generally decided against overloading the main body of the text.

Expanding figure 7 to 3 panels would generally be possible, if required, but would also reduce its readability. The same applies to a related overview table for n and k values.

To nevertheless provide an option for a cross-treatment comparison, we would like to suggest providing the suggested overview figure and table in the supplements, while referring to them in the main text. We are open for all suggested options.

**L242: Please specify what is meant by "some treatments".**

We will make it explicit in the new version of the manuscript.

**L279: What is meant by "in dependence to a variable density"?**

Since the density for each individual particle is unknown, a range of densities was assumed to estimate a realistic range for sinking velocities (shown as the gray range, centered in Fig. 8). Also see the reply for the next comment.

L271-285: I found it hard to follow this paragraph and to understand when particles observed by SEM, particles measured by FlowCam, and purely calculated values are referred to.

This section was reworked as:

**[New]:**

The gravitational sinking velocities of precipitated particles were measured using a FlowCam setup (see Bach et al.,2012 for technical details). Based on the concept of equivalent spherical diameters (ESD) the density of each particle was calculated, revealing a range from 1.54 to 3.18 g cm-3 for ESD sizes between 12 and 50  $\mu$ m. The average density was determined to be 2.358 g cm-3. The discrepancy with the density of aragonite (~2.95 g cm-3) may result from an overestimation of particle sizes in the calculation method, which relies on an inversion of Stokes' Law for the terminal sinking velocity of perfect spheres. However, most particles are non-spherical and contain numerous internal cavities within their structure (see Fig. S1), and their densities are therefore expected to be lower than those of pure aragonite. The determined particle density was then used to calculate the theoretical sinking velocities of the manually counted precipitated particles. To account for potential variability in particle density, Fig. 8 presents a range of sinking velocities of the counted precipitates.

Measured sinking velocities for precipitated particles within the aforementioned density range varied from  $^{-5}$  m d $^{-1}$  (14  $\mu$ m particle) to  $^{-47}$  m d $^{-1}$  (41  $\mu$ m particle). Recorded particles in the ESD range of 50-180  $\mu$ m were not included in the calculations, as they were not observed within the same filter material that was analyzed by visual inspection. Discrepancies between measured and calculated values may reflect aggregation effects or technical limitations of the utilized FlowCam to track particles smaller than 3  $\mu$ m (Bach et al., 2012).

**[Old]:**

Based on the distributions of equivalent spherical diameters (ESD), the sinking velocities of the precipitated materials were calculated to identify their hypothetical sinking velocities. To facilitate this calculation, the densities of the aragonite precipitates were determined by actual sinking velocity measurements of the same materials, providing densities of 1.54 to 3.18 g cm-3 in an ESD range of 12-50  $\mu$ m. The discrepancy with the density of aragonite (~2.95 g cm-3) may result from an overestimation of particle sizes in the calculation method, which relies on an inversion of Stokes' Law for the terminal sinking velocity of perfect spheres. However, most particles are not spherical and contain numerous cavities within their structure, which likely contributes to an underestimation of particle densities.

Therefore, Fig. 8 features a range of sinking velocities of the counted precipitates in dependence to a variable density, supported by ESD distributions and ranges for different types of precipitated particles. Measured sinking velocities for precipitated particles within the aforementioned density range varied from  $^{-5}$  m d $^{-1}$  (14  $\mu$ m particle) to  $^{-47}$  m d $^{-1}$  (41  $\mu$ m particle). Recorded particles in the ESD range of 50-180  $\mu$ m were not included in the calculations, as they were not observed within the same filter material that was analyzed by visual inspection, yielding densities of 1.1-1.3 g cm $^{-3}$ . Discrepancies between measured and calculated values may reflect aggregation effects for very high values and the technical limitations of the utilized FlowCam to track particles smaller than 3  $\mu$ m (Bach et al., 2012).

Figure 8: This figure is very busy and should be split up into multiple panels. The particle size distributions should be presented on their own with a clear x-axis. The y-axis is not easily understandable (does it represent both depth and sinking velocity, or depth divided by sinking velocity?), and I do not see what it is related to. Finally, the aragonite density is three orders of magnitude too high.

The authors agree that the density of the information presented is somewhat overwhelming as multiple aspects are shown, 1. sinking velocities, 2. particle size distributions and 3. residence times. Plotting the information in one figure allows the datasets to communicate with each other, as their information is complementary. To enhance readability, we will follow your suggestion and move the particle size distribution to a separate panel. Another y-axis to divide depth and sinking velocity will be added as well. The unit for densities will be corrected, thanks a lot for noticing.

Section 4.1: As I outlined in my main comment, the current model is not predictable, except within the same environmental conditions. Since the model is not actually linked to environmental conditions, it will also not be possible to implement it in ocean models.

We agree that the presented concepts are not generally applicable. L297-298: "For the present study, the compiled concepts allowed the description of the principles guiding the entire runaway process in site-specific condition". We agree that the presented concepts should be enhanced by a functional term for temperature, salinity, and particle density. With ongoing and future experiments, we are aiming to implement variable environmental conditions. Since this is part of a multi-year project, with each experiment lasting approximately three months, additional time is needed to collect the data density required for such a parameterization.

L316: Whitings are precipitation events, not a cause of precipitation.

Wording will be adjusted.

L392: "section", not "chapter".

This will be adjusted.

L399-400: Yes, and this should be reflected in the logistic equations.

Related sentence L399-400: "The interplay of precipitation potential by  $\Omega_{ar}$  and the practical available surface area could therefore be determined as the primary factors guiding the actual observed TA-loss rates."

Thank you for the comment. Based on your suggestion the calculations for Fig. 9 and S8 were refined. For each panel a new graph was added, which represents the empirical rate equation  $[r = k A (\Omega_{ar} - 1)^n]$

including the determined values for n and k; a continuous function for the surface area A (derived from the curve-fitted logistic TA-loss function) and another continuous curve-fitted logistic function for  $\Omega_{ar}$ . All these parameters combined provide an estimate of the realized aragonite precipitation rate (blue dotted graph). Here is an example plot for  $\Delta TA_{2000}$ . Hollow blue markers thereby represent the rate equation with the original  $\Omega_{ar}$  values.

Additionally, the mentioned continuous surface area function (red - dash-dotted graph) replaced the previously reported raw surface area values, which are still given by the red diamonds. The shown diagram represents an example plot as a replacement for the  $\Delta TA_{2000}$  panel in Fig. S8 (which will be labeled (a)-(f)). All other diagrams of the same type will be adjusted accordingly (Fig. 9 and remaining panels in Fig. S8)

L402-406: But k and n were derived from experiments in natural seawater, containing inhibitors, and based on the experimentally determined saturation state of aragonite (which is then used for the calculation of R). As such, these constants should include the potential impact of inhibitors. Is it not more likely that issues with accurately determining PSA are causing the difference?

L402-406: "For comparison, the blue data points in Fig. 9 represent the calculated theoretical loss rates at each sampling day, by inserting the experimentally determined  $\Omega$ ar and PSA values into the related empirical rate equation for  $\Delta$ TA2000 (see Tab. 3). As this equation does not account for any inhibitory factors, the resulting rates exhibit a slight positive bias compared to the observed values." With the new plotting approach (see above), the described effect will be less obvious visually. The text will nevertheless be adjusted accordingly, including a statement for the possible influence of PSA.

L430-434: Again, this shows that temperature and salinity need to be considered in the logistic equation.

L430-434: "Significantly shorter induction times were identified for subtropical conditions (Temp.  $^23^{\circ}$ C, Sal.  $^36$  psu, TA  $^2400$  µmol kg $^{-1}$ ). Hartmann et al. (2023) described an onset of the precipitation after just 4 days for a 50µm filtered neq incubation with initial values of 1050 µmol kg $^{-1}$  for  $\Delta$ TA and  $^15$  for  $\Omega_{ar}$ ."

We appreciate the suggestion. To our knowledge, all attempts to incorporate terms for temperature and/or salinity to reliably describe carbonate precipitation in natural seawater have so far not been successful. As a result, we are currently conducting new sets of experiments on Gran Canaria to obtain the data necessary to extend the function with temperature sensitivity. Through the ongoing and planned future experiments, we will aim to derive such an equation, considering the diverse environmental parameters that influence precipitation behavior (also see reply to the general comment above).